# CREDIT-BUDGETED ICPC-STYLE CODING: WHEN AGENTS MUST PAY FOR EVERY DECISION

**Lingfeng Zhou**[1*]**, Junhao Shi**[1*]**, Jin Gao**[1]**, Dequan Wang**[1,2†]

[1]Shanghai Jiao Tong University    [2]Shanghai Innovation Institute

## ABSTRACT

Current evaluations of autonomous coding agents assume an unrealistic, infinite-resource environment. However, real-world software engineering is a resource-bound competition. As we scale toward large agent swarms, ignoring compute and time costs risks catastrophic budget exhaustion. To shift the focus from isolated accuracy to cost-aware problem-solving, we introduce USACOArena[1], an interactive ACM-ICPC-style arena driven by a strict "credit" economy. Every generated token, local test, and elapsed second depletes a fixed budget, forcing agents to make strategic trade-offs. Our comprehensive profiling reveals that frontier single agents and swarms currently fail to optimally balance accuracy with these constraints, exhibiting divergent, path-dependent behaviors. Ultimately, US-ACOArena provides an essential dynamic training ground for developing highly efficient, resource-aware agent architectures.

## 1 INTRODUCTION

Autonomous agents like Claude Code and Codex have become highly capable programmers, routinely solving complex coding tasks. However, this rapid progress masks a critical flaw: current evaluations focus almost exclusively on static coding accuracy, implicitly assuming an idealized, infinite-resource environment. In reality, software engineering is a resource-bound competition. As the field scales toward deploying large agent swarms, ignoring opportunity costs—such as API tokens, execution time, and local testing compute—will lead to catastrophic resource exhaustion (Figure 1). Therefore, evaluating true agentic intelligence requires a fundamental paradigm shift: from isolated code correctness to competitive, cost-aware resource management.

To bridge this gap, agents require a dynamic feedback signal that quantifies the real-world penalty of their actions. We argue for the necessity of a generalized "credit" system—a unified budget encompassing every decision an agent makes. Under this framework, agents must pay for every generated token, every local compilation, and every passing second. This scarcity-driven competition forces agents to make strategic, metacognitive trade-offs: they must constantly evaluate whether to expend their depleting resources exploring a novel solution path or to halt and submit the current best attempt. Fostering this rigorous resource awareness in single agents today is an essential prerequisite for ensuring that future agent swarms can collaborate efficiently without exhausting shared budgets.

Accurately measuring an agent's ability to balance these constraints requires a rigorous, frictionless testing environment. Real-world software engineering benchmarks (Jimenez et al., 2023; Yang et al., 2024; 2025a; Jain et al., 2025b; Zhou et al., 2025; Badertdinov et al., 2025) are invaluable, but they inherently introduce confounding variables—such as missing documentation, ambiguous requirements, or complex repository setups—that obscure the pure observation of decision-making costs. To isolate and evaluate the resource-allocation capabilities of agents, we turn to the ACM-ICPC competitive programming format. Competitive programming inherently models a self-contained game with perfect information, where the rules, objective metrics, and strict computational budgets are fully disclosed upfront.

---

*Equal contribution    †Corresponding author: dequanwang@sjtu.edu.cn

[1]https://github.com/maple-zhou/USACOArena

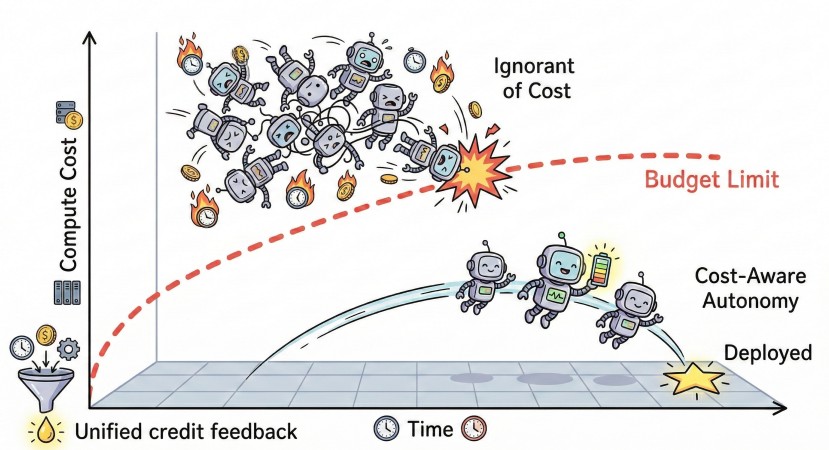

Figure 1: From Temporal Vacuum to Cost-Aware Autonomy. Without economic feedback, traditional swarms waste time and compute, inevitably crashing into real-world Budget Limits (top). USACOArena solves this by translating absolute time, tokens, and testing overhead into a Unified Credit Feedback signal (bottom-left funnel). By forcing swarms to "pay" for every decision, it induces efficient, cost-aware coordination for successful deployment (bottom right).

Based on these principles, we introduce USACOArena, an interactive coding arena driven by a strict credit economy. In USACOArena, every action consumes credits from a fixed, depleting pool. This framework translates a static coding test into a dynamic resource management survival challenge. It compels agents to utilize their remaining budget as a continuous feedback signal, pushing them to optimize the delicate balance between speed, cost, and accuracy.

We conduct comprehensive experiments in USACOArena to evaluate how current frontier models handle these economic and competitive constraints. Our empirical analysis yields three key insights. First, we benchmark a variety of leading single-model agents and profile their decision-making processes, uncovering distinctly divergent strategies: for instance, Gemini-2.5-pro exhibits an aggressive, high-exploration approach, whereas GPT-5-Codex demonstrates a highly conservative resource management profile. Second, we conduct self-play experiments to observe emergent behaviors. We find that identical agents, when operating under the same environmental pressures, discover highly diverse and path-dependent routes to solve the identical problem. Finally, by evaluating contemporary agent frameworks and early swarm implementations, we discover that currently, neither single agents nor agent swarms can optimally balance coding accuracy with strict credit limits. These results underscore a fundamental gap in current agent architectures, demonstrating that both single agents and collaborative frameworks currently fail to optimally manage resources.

In summary, our contributions are as follows:

1. A Shift to Credit Feedback: We identify the lack of resource awareness as a major bottleneck for the real-world deployment of agents and swarms. We shift the evaluation focus from simple code correctness to dynamic, cost-effective problem-solving under competitive constraints.

2. The USACOArena Arena: We present an interactive, ICPC-style environment driven by a strict credit budget. It rigorously enforces constraints by translating API tokens, local test limits, and elapsed time into a unified, quantifiable feedback metric.

3. Strategic Profiling of Frontier Models: We provide a comprehensive empirical analysis of how leading single agents (e.g., Gemini-2.5-pro, GPT-5-Codex) handle the trade-offs between speed, cost, and correctness, highlighting their disparate strategies and current limitations.

4. Pathways for Future Swarms: Through self-play and multi-agent experiments, we demonstrate that a strict credit economy elicits complex, path-dependent behaviors. USACOArena serves as a vital dynamic training ground to build highly efficient and resource-aware agent swarms in the future.

## 2 RELATED WORK

Our research is positioned at the intersection of automated code generation and the rapid emergence of autonomous agent swarms. We introduce a novel evaluation paradigm that bridges a critical gap: shifting the focus from static code correctness to cost-aware autonomy and strategic resource management under strict economic feedback.

### 2.1 LLMs FOR CODE AND THE ILLUSION OF STATIC BENCHMARKS

Recent breakthroughs in Large Language Models (LLMs) have significantly advanced automated code generation (OpenAI, 2025b; Anthropic, 2025a;b; Guo et al., 2024; Wei et al., 2024; DeepSeek-AI et al., 2024; Cummins et al., 2024; Liu et al., 2024). To assess these capabilities, benchmarks evolved from fundamental functional correctness (Chen et al., 2021; Austin et al., 2021) to algorithmic competitions (Li et al., 2022; Shi et al., 2024) and real-world software engineering tasks (Jimenez et al., 2023; Li et al., 2025; Liu et al., 2025). Recent efforts further enhanced evaluation rigor with live problem sets (Jain et al., 2024; Zheng et al., 2025), robust ranking systems (Quan et al., 2025; Yang et al., 2025b), multiple function calls (Zhuo et al., 2025), language-driven coding (Deng et al., 2025), and interactive debugging (Yuan et al., 2025). However, these benchmarks share a fundamental limitation: they evaluate models in a temporal vacuum. By focusing exclusively on the static accuracy of the final code output, they fail to measure the economic realities of software engineering, leaving factors like compute cost, temporal efficiency, and strategic trade-offs completely unevaluated.

### 2.2 FROM INDIVIDUAL AGENTS TO CODING SWARMS

To move beyond simple code generation and drastically compress delivery time, the community is rapidly pivoting toward coordinated multi-agent swarms. Sophisticated frameworks feature diverse architectures, including coder-tester co-evolution (Wang et al., 2025b), generator-verifier pairs (Jain et al., 2025a), policy-critic models (Xie et al., 2025), and retriever-generator pipelines (Wang et al., 2025a), alongside internal reasoning mechanisms such as feedback loops and explanation-driven repair (Jiang et al., 2025; Gehring et al., 2025). Additionally, multi-agent systems built on platforms like AutoGPT (Significant Gravitas), MetaGPT (Hong et al., 2024), AgentVerse (Chen et al., 2024b), and OpenHands (Wang et al., 2024; Ma et al., 2024; Gao et al., 2024; Chen et al., 2024a; Yang et al., 2024; Xia et al., 2024; Aggarwal et al., 2025) employ complex workflows to parallelize intelligence. Despite this architectural complexity, swarm effectiveness is still judged using static benchmarks. This creates a severe disconnect: a swarm might successfully solve a task but waste massive amounts of compute or succumb to heavy coordination overhead. The true strategic competence of these swarms remains largely unquantified.

### 2.3 THE MISSING PARADIGM: ECONOMIC FEEDBACK AND COST-AWARE AUTONOMY

In Reinforcement Learning, learning from experience in interactive environments is crucial for achieving superhuman performance. Works like SWE-rebench (Badertdinov et al., 2025) and R2E-Gym (Jain et al., 2025b) provide executable training grounds, while evaluation frameworks like ColBench (Zhou et al., 2025) and TheAgentCompany (Xu et al., 2025) measure multi-step task performance. However, they primarily serve as execution sandboxes that ultimately reward only the final artifact.

In contrast, USACOArena is explicitly designed to instill procedural wisdom through direct economic feedback. By theoretically framing the arena as a Budget-Constrained Partially Observable Markov Decision Process (POMDP), we force swarms to pay for every decision, inference token, and elapsed second. While existing benchmarks measure *Crystallized Intelligence* (applying known patterns), USACOArena rigorously tests *Fluid Intelligence*. This isolates a swarm's ability to optimize algorithmic reasoning, manage risk aversion, and mitigate coordination tax under strict uncertainty to achieve true cost-aware autonomy.

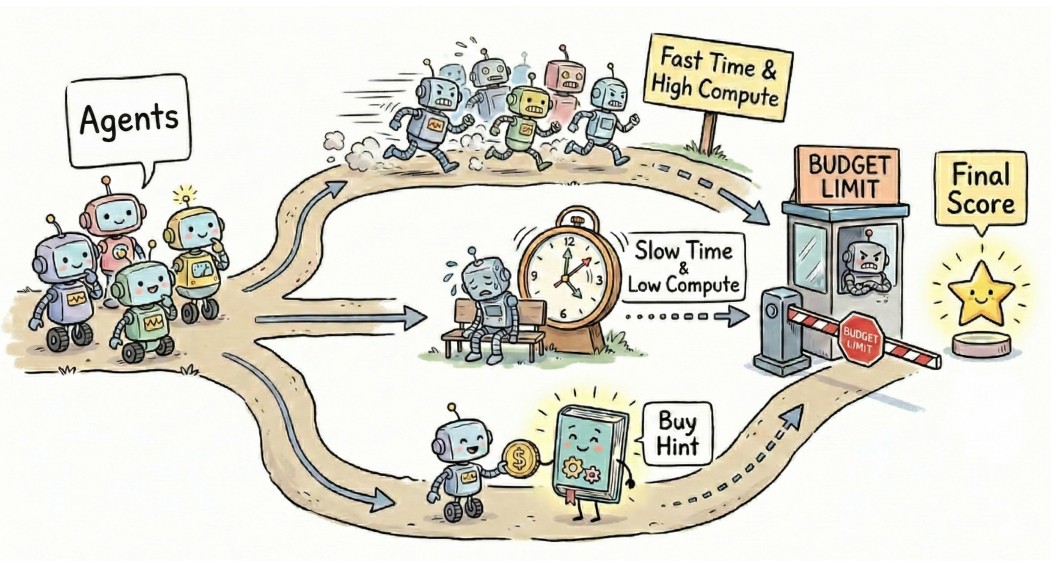

Figure 2: The Unified Credit Economy of USACOArena. Our environment evaluates agents on cost-aware decision-making rather than static correctness. The agents face distinct strategic trade-offs: aggressively parallelizing to minimize absolute delivery time at the expense of high inference costs (top) , conserving compute while burning valuable time credit (middle) , or spending resources to acquire strategic hints to overcome bottlenecks (bottom). Every action—along with the continuous passage of wall-clock time—drains a shared Budget Limit. To maximize the Final Score , the swarm must optimally balance speed, compute, and reliability without going bankrupt.

## 3    USACOARENA: AN ICPC-INSPIRED ARENA FOR CODING AGENTS

This section details the design of USACOArena. It is an interactive environment built to evaluate the strategic choices of coding agent swarms. We move beyond simple correctness checks. We aim to evaluate how well agents manage delivery time and compute costs. First, we explain our ACM-ICPC-style design and the USACO problem set (Section 3.1). Next, we define our unified credit model (Section 3.2). This model unifies delivery time, inference tokens, and testing overhead into a single credit economy. Finally, we explain our system architecture and communication protocol (Section 3.3).

### 3.1    FOUNDATIONAL DESIGN: ADAPTING THE ACM-ICPC FORMAT

We model USACOArena on the ACM-ICPC format. This format matches real-world software engineering well. The ACM-ICPC rules use an all-or-nothing scoring system. This rewards fast and bug-free code generation. This differs from the IOI format, which focuses on theoretical limits. Appendix C provides more details.

We make two main changes for agent swarms. First, official ACM-ICPC problems are too hard for current models. We use the USACO problem corpus instead. Its difficulty ranges from Bronze to Platinum. This allows us to test different agent capabilities clearly. Each match uses 12 problems to simulate an ACM-ICPC World Final.

**Mitigating Data Contamination.**    To prevent train-test overlap, we use a "Living Benchmark" approach. USACO releases four new contests every year. We update our dataset with the newest season. This tests the agents on new, unseen problems.

Second, we add absolute delivery time into a unified credit economy. In the real world, fast delivery is as important as correct code. We give each agent a fixed credit budget. Actions and the passing of time both cost credits. This forces agents to balance speed, compute cost, and reliability. To join the main test, an agent must first solve a simple Bronze problem (Section 4.1).

The agent's policy $\pi$ outputs a tuple: (Score, Consumed Credit). The agent must maximize its score without exceeding the credit limit:

$$\max_{\pi} \text{Score}(\pi) \quad \text{subject to } C_{\text{action}}(\pi) + C_{\text{time}}(\pi) \leq C_{\text{limit}} \tag{1}$$

## 3.2 Scoring and Credit Model: Operationalizing ACM-ICPC Rules

To implement these ideas, USACOArena uses an explicit scoring and credit system. This system mirrors the real-world economic pressures of software engineering.

**Ranking and Scoring.** As shown in Figure 2, we adapt the ACM-ICPC rules. Total score is the primary ranking metric. Consumed credit is the tie-breaker. The score is a weighted sum of all accepted (AC) problems. Harder problems give more points. We give no partial credit for failing any test case.

**The Unified Credit Model.** To implement these ideas, USACOArena uses an explicit scoring and credit system. This system mirrors the real-world economic pressures of software engineering.

**Ranking and Scoring.** As shown in Figure 2, we adapt the ACM-ICPC rules. Total score is the primary ranking metric. Consumed credit is the tie-breaker. The score is a weighted sum of all accepted (AC) problems. Harder problems give more points. We give no partial credit for failing any test case.

**The Unified Credit Model.** The credit system is a unified economy. It includes action costs, delivery time costs, and penalties.

Action Costs are the resources spent to solve problems.

- **LLM Inference Cost:** This is the compute cost. We normalize it by the API price (Appendix H). Expensive models use more credit. Agents must balance thinking quality and cost.

- **Hint and Test Cost:** These represent the effort for reading documents or local debugging (Appendix K).

Delivery Time Cost is our key addition to evaluate speed. We track the absolute wall-clock time $T$ from the start to the end of the agent's run. We multiply $T$ by an adjustable time coefficient $\alpha$. This converts absolute delivery time into credit. By changing $\alpha$, we can test different competition scenarios. A high $\alpha$ values fast delivery, while a low $\alpha$ values compute efficiency.

Penalty Costs happen when an agent submits wrong code (e.g., Wrong Answer). This stops agents from guessing randomly.The termination rule is important. An agent stops only when its action and time costs exceed the budget ($C_{\text{action}} + \alpha T \leq C_{\text{limit}}$). However, the final tie-breaker uses the total consumed credit. This includes penalties ($C_{\text{consumed}} = C_{\text{action}} + \alpha T + C_{\text{penalty}}$). This forces agents to deliver correct code quickly and cheaply.

## 3.3 System Architecture and Communication Protocol

We build our system using a turn-based loop based on the Model Context Protocol (MCP) (Anthropic, 2024). Every turn, the USACOArena server sends the game state to the agent as a JSON object. This includes credit, score, and the leaderboard. The agent replies with an action, like SUBMIT_SOLUTION. This protocol ensures all agents see the same data and use the same actions. We run all submitted code in a secure online judge sandbox. This makes our platform safe and easy to reproduce.

We choose MCP to solve the reproducibility crisis in agent research. MCP is independent of programming languages. Researchers can test any agent easily. It also supports multi-turn chats. This helps us build future multi-agent ACM-ICPC teams easily.

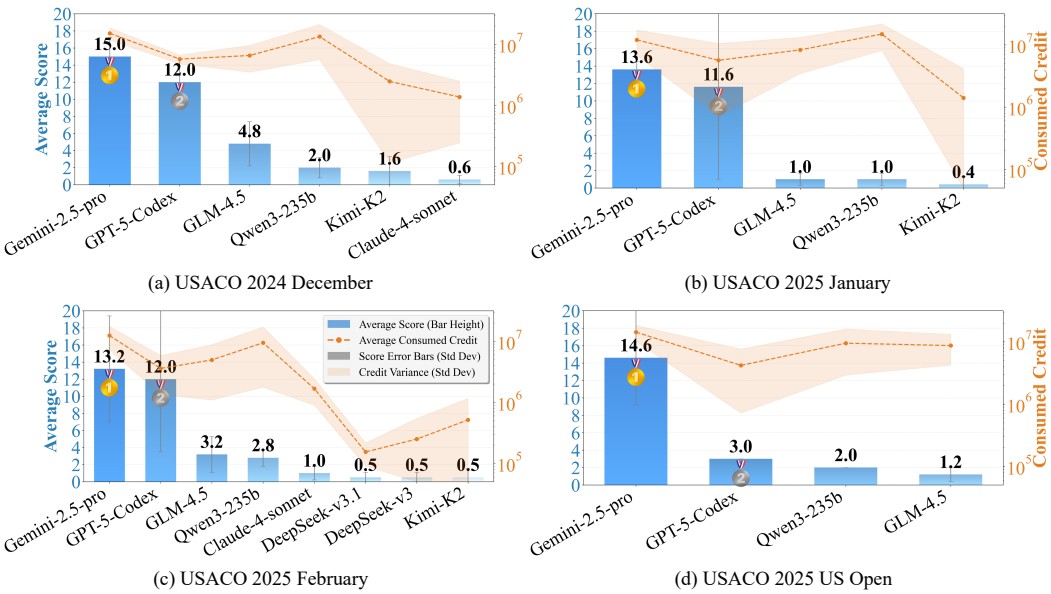

Figure 3: **Average agent scores and consumed credit across the four contests of the 2024–2025 USACO season.** Each subplot shows the results for a single contest, with agents sorted by rank. Blue bars represent the average score (left axis), while the orange line indicates the average consumed credit (right axis, log scale). Error bars and the shaded area denote the standard deviation over five independent runs; for clarity, only agents that achieved a non-zero average score are shown. The results reveal a stable and significant performance hierarchy: across all four contests of varying difficulty, Gemini-2.5-pro and GPT-5-Codex consistently rank first and second, respectively.

## 4 EXPERIMENTS

Our evaluation demonstrates the utility of USACOArena. We first detail the experimental setup (Section 4.1). Next, we present the main competition results across diverse LLMs (Section 4.2). We explore emergent behaviors in self-play (Section 4.3). Furthermore, we conduct systematic ablations of our benchmark (Section 4.4). Finally, we isolate the impact of agent swarm and evaluate absolute delivery time using Codex (Section 4.5).

### 4.1 EXPERIMENTAL SETUP

All experiments follow the ICPC-inspired competition rules detailed in Section 3.

**Problem Corpus and Qualification.** The evaluation uses 48 problems from the 2024–2025 US-ACO season. To ensure baseline competency, an agent must successfully solve the easiest Bronze-level problem from a contest to qualify. Detailed difficulty baselining and qualification results are provided in Appendix D and E.

**Agent Construction.** Agents are constructed by pairing an LLM with a non-prescriptive prompt outlining the objective rules, ensuring strategies are emergent rather than hard-coded (Appendix J).

### 4.2 CROSS-MODEL BENCHMARKING: STRATEGIC PROFILES UNDER STRICT COMPUTE BUDGETS

While Section 4.5 demonstrates the critical role of absolute delivery time in agents, evaluating diverse commercial APIs introduces unfair network latency. Therefore, to isolate and analyze the pure strategic reasoning capabilities of different foundation models, we temporarily set $\alpha = 0$ and focus entirely on compute efficiency. To ensure the robustness and generalizability, we evaluate agents across the four distinct contests of the 2024–2025 USACO season. Each contest is run five times,

and the results presented in Figure 3 are the average of those runs. For this experiment, we select GPT-5-Codex as the representative for the GPT-5 series, as it is specifically optimized for agentic coding tasks (OpenAI, 2025a).

**A Stable Two-Tier Hierarchy.** The results reveal a consistent two-tier performance hierarchy across all four contests. Gemini-2.5-pro and GPT-5-Codex invariably secure the first and second ranks, respectively, demonstrating a significant and reliable capability gap between these top-tier agents and the rest of the field, whose performance is far more volatile. For clarity, the figure only displays agents that achieve a non-zero average score in each contest.

**In-Depth Analysis of Top-Tier Agents.** While the main competition results establish a clear performance hierarchy, they do not fully explain *why* one agent consistently outperforms another. The final scores are merely the outcome of a complex, dynamic decision-making process. To understand the underlying strategic differences that lead to victory, this section provides an in-depth analysis of the two top-performing agents: Gemini-2.5-pro and GPT-5-Codex.

**Capability vs. Strategy Gap and Human Baselines.** A critical observation is that the benchmark is far from saturated. The theoretical maximum score per contest is 54 points (weighted across Bronze to Platinum levels). Currently, top-tier agents score around 15 points. To contextualize this against human baselines, we map agent scores to the official USACO division structure:

- **Novice (Score $< 3$, Bronze):** Struggling to clear entry-level algorithmic tasks.
- **Intermediate (Score 3–9, Silver):** Capable of solving foundational problems.
- **Advanced (Score 9–24, Gold):** Current state-of-the-art agents (Score $\sim 15$) fall firmly into this tier. They are comparable to talented high school competitors who have cleared intermediate stages but struggle with complex algorithmic reasoning.
- **Expert (Score $> 24$, Platinum):** The world-finals frontier that current models have yet to consistently breach.

Remarkably, GPT-5-Codex has occasionally achieved peak scores of up to 29 in isolated runs, proving the intrinsic capability to solve Platinum problems exists. The bottleneck keeping its average score much lower is purely strategic—a failure to optimally allocate budget and manage risk.

This analysis resolves the performance paradox: Gemini-2.5-pro wins by being a more effective competitor, not necessarily a superior problem-solver. This distinction is best understood through the exploration-exploitation trade-off. Gemini-2.5-pro's strategy is one of aggressive exploration; it attempts many problems to maximize broad coverage and its cumulative score, accepting a lower precision rate as a necessary cost. In contrast, GPT-5-Codex's cautious perfectionism is a form of pure exploitation. Its risk-averse approach limits its attempts to only high-confidence problems, causing it to miss many scoring opportunities and turning its precision into a strategic liability.

This distinction highlights a key finding of our work. Gemini-2.5-pro's success demonstrates that in a competitive setting, a strategy of broad exploration can outperform a more capable but overly conservative exploitation strategy. This implies that optimal performance in a complex, resource-constrained environment like USACOArena requires more than just raw problem-solving accuracy. For the field to advance, this suggests that developing an agent's decision-making framework—its ability to assess risk and manage resources—is as important as enhancing its core capabilities. True expertise in this domain lies in an agent's ability to dynamically balance the trade-off between exploring all viable opportunities and exploiting the most promising ones.

**Impact of Contest Difficulty.** The varying difficulty of the contests highlights the extent of this performance gap. While more agents achieve non-zero scores in accessible contests, the most challenging one—the USACO 2025 US Open–showcases a dominant performance by Gemini-2.5-pro. Its score of 14.6 establishes a vast lead over GPT-5-Codex (3.0), suggesting that high-difficulty problems strongly accentuate the top agent's strengths. A detailed breakdown is in Appendix F.

**Limitations in Agent Self-Assessment.** However, a deeper analysis of agent behavior reveals a widespread deficiency in strategic self-assessment. Lower-ranked agents often mismanage resources

A deeper look at the data reveals a fascinating paradox. As summarized in Table 1, GPT-5-Codex demonstrates a significantly higher peak capability, achieving a maximum score of 29 compared to Gemini-2.5-pro's 19. This confirms its potential to solve more difficult, higher-value problems. However, it is Gemini-2.5-pro that achieves a better average rank and a win rate more than double that of its competitor (68.4% vs. 31.6%). This performance inversion points directly to the decisive role of competitive strategy.

The strategic profiles in Figure 4 explain this outcome, revealing a clear difference. Gemini-2.5-pro's profile is characterized by an aggressive, high-volume strategy. Its plot extends outward on the **Attempted Problems** and **Submission Counts** axes, indicating it attempts more problems to maximize scoring opportunities. This "breadth-first" approach treats credit as a resource to be actively spent in exchange for broader coverage.

In stark contrast, GPT-5-Codex adopts a conservative, "perfectionist" strategy. Its profile is heavily skewed toward near-perfect **First-Submit Accuracy** and **Problems Solve Rate**. This risk-averse approach prioritizes precision, but severely limits its problem coverage, causing it to forego attempts on many potentially solvable problems.

Table 1: Comparison of Agent Profile Metrics for Gemini-2.5-pro and GPT-5-Codex.

| Agent | Avg. Rank | Win Rate | Max Score | Min Score |
|---|---|---|---|---|
| Gemini-2.5-pro | $1.3 \pm 0.47$ | 70.0% | 19 | 4 |
| GPT-5-Codex | $1.7 \pm 0.47$ | 30.0% | 29 | 3 |

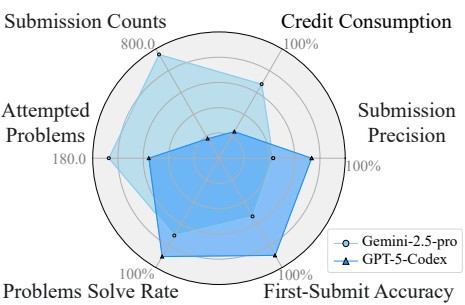

Figure 4: **Strategic Profiles of Top-Tier Agents.** Submission Precision is the percentage of AC submissions out of all submission attempts; Problems Solve Rate is the percentage of AC problems out of all attempted problems; and First-Submit Accuracy is the percentage of problems solved on the first attempt out of all successfully solved problems.

by attempting problems far beyond their capabilities, forgoing points on easier tasks. Paradoxically, the top-performing agent exhibits the opposite flaw: despite being capable of solving high-value Platinum problems (see Appendix D), Gemini-2.5-pro consistently defaults to safer, lower-scoring problems. This suggests that even the most advanced agents currently lack sophisticated risk-assessment and strategic planning.

Further ablation studies within the GPT-5 family (detailed in Appendix G) confirm that agentic frameworks significantly improve performance precision, albeit at higher resource consumption.

### 4.3 EMERGENT BEHAVIORAL DIVERSITY IN SELF-PLAY

To investigate whether a top agent's performance is deterministic, we conduct a series of self-play experiments, pitting two identical instances of `gemini-2.5-pro` against each other. The results reveal a striking diversity of behaviors. As shown in Figure 5(a), the outcomes across 18 competitors are highly variable, rarely ending in a tie, and showing no simple correlation between credit consumed and final score. A trajectory analysis of a single match (Figure 5(b)) provides a granular explanation for this variance. It shows how different, path-dependent strategic choices—such as a methodical, bottom-up approach versus getting stuck on difficult problems—can lead to a decisive win-loss outcome.

These findings yield a key insight: USACOArena is not a simple puzzle but a complex and sensitive environment that can reveal critical inconsistencies in an agent's decision-making. This demonstrated behavioral diversity, where a single policy can produce a wide range of outcomes, is a crucial prerequisite for improvement through learning-based methods. Therefore, our self-play experiments validate USACOArena not only as a robust evaluation testbed but also suggest its potential as a dynamic training ground for future research into cultivating more capable agents.

A granular trajectory analysis (Figure 5(b)) reveals the underlying mechanics of this variance, which we term the **First-Move Effect**. Divergence is highly correlated with the outcome of the agent's first attempted problem. If the agent stumbles upon a solvable problem early, it enters a *Success Loop*,

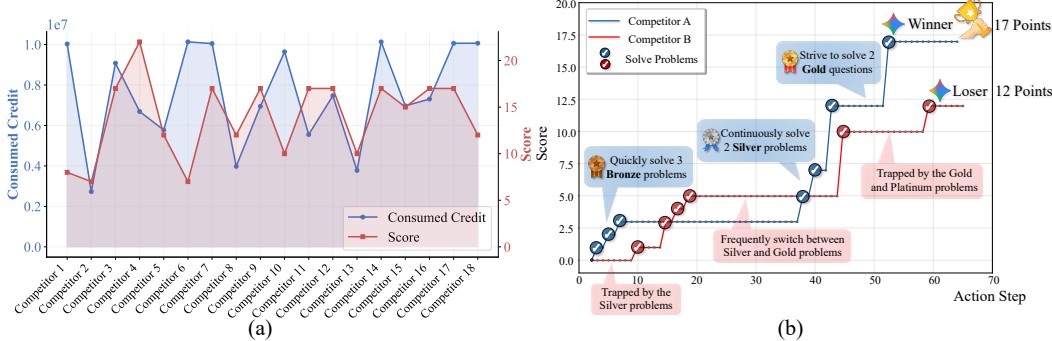

Figure 5: **Emergent behavioral diversity and strategic divergence in self-play.** (a) Final scores and credit consumed across nine competitions between identical `gemini-2.5-pro` agents, revealing a wide spectrum of outcomes with no trivial correlation between cost and performance. (b) A trajectory analysis of a single match provides a granular explanation, showing how different strategic paths lead to a decisive win-loss result. This demonstrated diversity, resulting from complex path-dependent decisions, validates USACOArena's suitability as a rich training environment.

gaining the "confidence" (score padding) to take calculated risks on harder problems. Conversely, an identical agent that randomly selects a mathematically intractable problem first often enters a *Panic Loop*: it fails, rapidly depletes its budget, and adopts an overly conservative fallback strategy, leading to a performance collapse. This rich, path-dependent divergence validates USACOArena as a powerful environment with sufficient reward gradients for future Reinforcement Learning (RL) based agent optimization.

### 4.4 SYSTEMATIC ABLATIONS AND PROMPT SENSITIVITY

To validate that USACOArena measures intrinsic strategic intelligence rather than artifacts of arbitrary hyperparameter tuning, we conducted an extensive ablation campaign across 8 models and 7 environmental dimensions on the 2025 February Contest (detailed results in Appendix B).

**Capability Saturation vs. Constraint Effect.** By systematically varying the Credit Limit, we observed a strict *Constraint Effect*: reducing the budget to 10M significantly degrades the top model's (Gemini-2.5-pro) performance (dropping from 13.2 to 8.3). However, doubling the budget to 40M yields no improvement (remaining at ∼13.0). This confirms that current state-of-the-art agents are **Capability-Limited, not Budget-Limited**. They hit a "reasoning wall" long before they hit the credit limit, proving the hierarchy is robust. Furthermore, weaker models exhibit a clear floor effect, scoring near 1.0 regardless of resource abundance.

**Prompt Sensitivity and Intrinsic Alignment.** We additionally tested whether simple prompt engineering could fix the strategic deficits of models like Gemini-2.5-pro. We evaluated four prompt variations, ranging from enforced Chain-of-Thought (P1.1) to explicit "Aggressive/Opportunistic" instructions (P2.1). Even the most aggressive prompt yielded only marginal score improvements (13.2 to 14.7), while overly verbose Few-Shot prompts actually degraded performance by consuming excessive token credits without improving decisions. This confirms that the observed "Conservative" versus "Aggressive" profiles are intrinsic to the models' alignment and architectural priors, not merely a product of prompt formulation.

### 4.5 PROFILING AGENT SWARM: THE ECONOMIC TRADE-OFFS OF COST-AWARE AUTONOMY

Beyond single-agent performance, we investigate the ultimate resource management challenge: scaling to agent swarms, where parallel execution drastically accelerates both problem-solving speed and budget depletion. To isolate the impact of multi-agent coordination strategies from underlying model variations, we use swarms of Codex agents to evaluate their efficiency under strict credit constraints. We configured the swarms with three distinct strategic profiles: (1) Speedy Spendthrift:

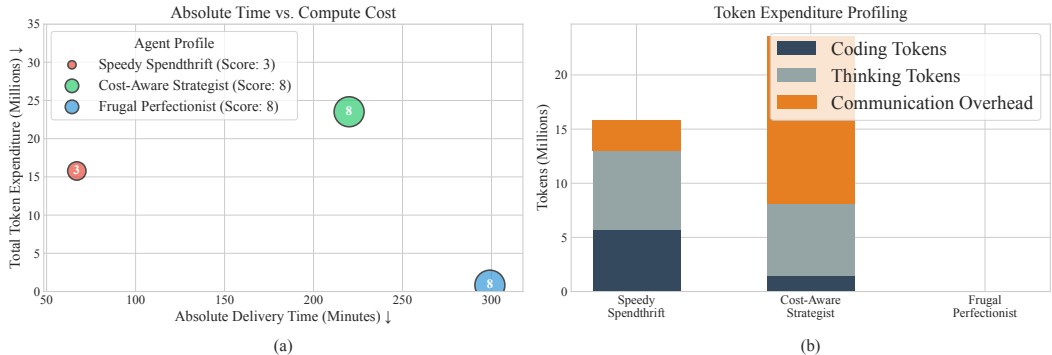

Figure 6: Performance and resource profiling of Codex agent swarms. (a) Absolute Time vs. Cost: Illustrates the trade-off between delivery time, token expenditure, and final score. The Speedy Spendthrift minimizes time but suffers a catastrophic performance drop (Score: 3), whereas the Frugal Perfectionist achieves a high score (Score: 8) at the cost of impractical delays ( 300 mins). (b) Token Breakdown: Reveals that blindly maximizing parallel workers (Speedy) triggers a massive explosion in Communication Overhead (orange hatched area). This rapidly depletes the budget without improving coding accuracy, empirically highlighting the critical need for dynamic, cost-aware resource management in future multi-agent systems.

Maximizes parallel workers to minimize absolute delivery time, disregarding token costs. (2) Frugal Perfectionist: Uses a cautious, sequential approach to minimize token expenditure. (3) Cost-Aware Strategist: Dynamically balances parallel exploration and sequential execution based on the remaining budget.

Figure 6(a) illustrates the fundamental trade-off between absolute delivery time and token expenditure. The *Speedy Spendthrift* effectively compresses delivery time but suffers from extreme token inflation. Conversely, the *Frugal Perfectionist* strictly conserves tokens at the cost of time delays.

Figure 6(b) reveals how resource mismanagement dictates final match performance under strict credit limits. Despite its speed, the *Speedy Spendthrift* achieves a poor score of 3. Its aggressive parallelization triggers a massive communication tax, prematurely draining its budget on inter-agent coordination rather than effective problem-solving. By dynamically interpreting economic feedback, the *Cost-Aware Strategist* mitigates this coordination overhead and efficiently allocates its budget. It successfully achieves an optimal winning score of 8 in a fraction of the time required by the sequential approach.

## 5 CONCLUSION

In this work, we introduce USACOArena, an interactive evaluation framework that translates absolute delivery time, inference tokens, and testing overhead into a unified credit economy. By forcing agents to pay for every decision, our experiments reveal that state-of-the-art performance relies less on raw coding capability and more on path-dependent strategies that carefully balance aggressive exploration with conservative precision.

However, our evaluation reveals critical limitations in current AI systems. Even top-tier models exhibit profound metacognitive deficits, frequently misallocating resources on intractable problems and ignoring strategic hints when trapped in failure loops. Furthermore, our ablation studies indicate that state-of-the-art agents are fundamentally capability-limited rather than budget-limited; even with abundant credits, they hit a "reasoning wall" and fail to translate surplus resources into higher scores. They also demonstrate a widespread lack of strategic self-assessment, oscillating between reckless gambles on advanced tasks and overly conservative choices despite being capable of more. Moving forward, the rich, non-deterministic behavioral divergence observed in our self-play experiments validates USACOArena as more than a static benchmark. It serves as a dynamic Reinforcement Learning training ground, providing the essential economic reward gradients needed to cultivate self-aware, cost-efficient agents.

## ACKNOWLEDGMENTS

This research is supported by the Key R&D Program of Shandong Province, China (2024CXGC010213). We express our gratitude to the funding agency for their support. We thank all the anonymous reviewers for their valuable suggestions.

## ETHICS STATEMENT

Our problem dataset is exclusively sourced from the publicly accessible USA Computing Olympiad (USACO) archives, which are distributed for educational and training purposes. Our usage falls strictly under non-commercial academic research, consistent with standard fair-use practices in the machine learning community. We do not redistribute the data for profit. The source code for the USACOArena environment, agent wrappers, and decision logs will be open-sourced to ensure full reproducibility.

## REPRODUCIBILITY STATEMENT

We are committed to ensuring the reproducibility of our work. The core methodology of our environment and the complete experimental setup are detailed in Section 3 and Section 4, respectively. The appendix provides exhaustive details necessary for reproduction, including: the full problem corpus from the 2024-2025 USACO season (Appendix D); comprehensive qualification and main competition results (Appendix E and F); a list of all large language models used (Appendix H); all competition hyperparameters (Appendix I); and the exact system prompt provided to the agents (Appendix J). The complete source code for the USACOArena environment, agent wrappers, and evaluation scripts will be made publicly available in an open-source repository upon publication.

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

## A  STATEMENT ON LLM USAGE

Throughout the preparation of this manuscript, we utilized a large language model (Google's Gemini) as a collaborative writing assistant. The model's contributions were primarily focused on the articulation and presentation of our research. Specific tasks included refining the paper's narrative structure and logical flow, condensing and polishing sentences for clarity and impact, suggesting alternative terminology to strengthen key arguments, and providing feedback on the design and captions of figures. The core scientific contributions—including the initial research ideation, the design and implementation of the USACOArena environment, the execution of experiments, and the analysis of the results—were conceived and conducted entirely by the human authors. The role of the LLM was that of a writing partner and editor. We are disclosing its use here to maintain full transparency regarding our research and writing process.

## B  SYSTEMATIC ABLATIONS AND ROBUSTNESS ANALYSIS

To address potential concerns regarding the sensitivity of our benchmark to specific hyperparameter choices, we conducted a comprehensive ablation campaign. We systematically varied the environment's Credit Limits, Economic Costs (Hints and Testing), and Incentive Structures (Scoring Weights) across all 8 models.

**Rationale for Problem Selection:** We executed these ablations on the USACO 2025 February Contest. Unlike the extremely difficult US Open or the easier earlier contests, the February Contest offers a balanced difficulty distribution (a "Goldilocks" zone). This allows most agents to solve early problems but forces them to make difficult, observable trade-offs on later ones, making behavioral shifts in decision-making highly apparent. Due to computational resource constraints, the ablation studies presented below are averaged over 3 independent runs.

### B.1  THE MEGA-ABLATION MATRIX

Table 2 presents the complete results of our sensitivity analysis across 7 environmental dimensions.

Table 2: The Mega-Ablation Matrix on USACO 2025 February Contest. The "Main Result" column reflects the standard configuration used in Section 4.2. "Exp. Score" uses exponential scoring weights {1, 10, 100, 1000} instead of the default {1, 2, 5, 10}.

| Model | Main Result (Table 10 config) | Low Credit (10M) | High Credit (40M) | Free Test ($0) | High Test ($1k) | Free Hint ($0) | High Hint (100×) | Flat Score | Exp. Score |
|---|---|---|---|---|---|---|---|---|---|
| Gemini-2.5-pro | **13.2** | **8.3** | **13.0** | **10.7** | **9.7** | **12.0** | **13.0** | **5.0** | **19.7** |
| GPT-5-Codex | 12.0 | 4.3 | 4.3 | 5.7 | 4.3 | 3.7 | 3.7 | 3.3 | 16.3 |
| GLM-4.5 | 3.2 | 1.0 | 0.7 | 1.0 | 1.0 | 1.0 | 1.7 | 1.3 | 1.0 |
| Qwen3-235B | 2.8 | 1.7 | 1.7 | 3.3 | 1.7 | 1.3 | 2.7 | 1.3 | 0.0 |
| Claude-4-Sonnet | 1.0 | 2.0 | 3.3 | 2.7 | 2.0 | 3.0 | 2.0 | 2.0 | 5.3 |
| DeepSeek-V3.1 | 0.5 | 0.7 | 1.0 | 1.0 | 0.3 | 1.0 | 1.0 | 0.7 | 1.0 |
| DeepSeek-V3 | 0.5 | 0.3 | 0.3 | 1.0 | 1.0 | 0.3 | 0.3 | 0.7 | 0.3 |
| Kimi-K2 | 0.5 | 1.3 | 1.7 | 1.0 | 0.7 | 1.0 | 1.0 | 1.3 | 1.0 |

### B.2  SCIENTIFIC INSIGHTS FROM THE ABLATION SPECTRUM

The extensive data from the Mega-Ablation Matrix confirms that the USACOArena evaluation framework successfully isolates *Strategic Intelligence* from environmental noise. The key scientific findings include:

- **Credit Limits (Constraint vs. Capability):** When reducing the budget to **10M (Low Credit)**, Gemini-2.5-pro's performance drops significantly (from 13.2 to 8.3). This proves that the credit budget is a highly meaningful constraint that accurately simulates real-world scarcity and forces necessary trade-offs. Conversely, doubling the budget to **40M (High Credit)** does not improve performance (13.2 to 13.0). This indicates that current state-of-the-art agents are *Capability-Limited*, not Budget-Limited. They hit a "reasoning wall" where throwing more computational resources (or money) at the problem yields sharply diminishing returns. Agents frequently terminate with a surplus of budget in the 40M setting, getting stuck in reasoning loops rather than effectively burning resources for progress.

- **Robustness of the Model Hierarchy:** The performance gap and overarching hierarchy (*Gemini-2.5-pro > GPT-5-Codex > Others*) remain invariant across drastic changes to Test Costs, Hint Costs, and Scoring Weights. This definitively proves that superior performance in USACOArena stems from intrinsic model capability and architectural fit, rather than overfitting to specific parameter tuning or exploiting arbitrary pricing structures.

- **The Floor Effect on Weaker Models:** Models with comparatively weaker coding or reasoning capabilities (e.g., DeepSeek, GLM, Kimi) hover near a score of $\sim 1.0$ regardless of the environmental settings (even with free hints or free tests). This validates that USACOArena effectively filters out models that lack the prerequisite *Fluid Intelligence* to even form a basic strategy. Only models that pass the initial capability threshold can participate in the deeper strategic meta-game measured by the arena.

## C    FOUNDATIONS IN COMPETITIVE PROGRAMMING STANDARDS

The design of USACOArena is deeply rooted in the well-established standards of human competitive programming. This appendix provides a detailed rationale for our two foundational design choices: (1) the adoption of a competitive programming format as the evaluation paradigm, and (2) the specific selection of the ACM-ICPC ruleset over other formats, such as the IOI. These choices are crucial for creating an evaluation framework that measures the capabilities most relevant to the practical application of coding agents in real-world software engineering contexts.

### C.1    THE RATIONALE FOR A COMPETITIVE EVALUATION PARADIGM

Current static benchmarks for code generation, such as HumanEval (Chen et al., 2021) and MBPP (Austin et al., 2021), primarily evaluate an agent's ability to produce functionally correct code for isolated, well-defined problems. While valuable, this static pass/fail approach fails to capture the dynamic, resource-constrained decision-making process that defines true autonomous agency. It assesses *what* an agent can produce, but not *how* or *why* it arrives at a solution.

For decades, programming competitions have served as the de facto standard for assessing human intelligence in computational problem-solving. They provide a holistic evaluation by creating an environment where participants must:

- **Manage Finite Resources:** Operate under strict constraints (e.g., time, computational resources), forcing strategic allocation of effort.

- **Prioritize Tasks:** Analyze a set of diverse problems, assess their difficulty, and strategically decide the order in which to tackle them.

- **Balance Trade-offs:** Make critical decisions between solution optimality, implementation speed, and correctness risk.

By situating agents in a competitive arena, we move beyond measuring mere correctness and begin to quantify these crucial agentic abilities. The competitive format transforms the evaluation from a simple test of knowledge into a rigorous assessment of strategy, efficiency, and performance under pressure, providing a far richer and more meaningful signal of an agent's true capabilities.

### C.2    WHY ACM-ICPC OVER IOI? A FRAMEWORK FOR ENGINEERING-ALIGNED EVALUATION

While both the ACM International Collegiate Programming Contest (ICPC) and the International Olympiad in Informatics (IOI) are premier competitions, their underlying philosophies and mechanics are tailored to measure different skills. We deliberately model USACOArena on the ACM-ICPC because its format is substantially more aligned with the values and demands of professional software engineering. The goal is to evaluate coding agents as potential engineering collaborators, not as pure research tools. This alignment is evident across several key dimensions.

**Evaluation Paradigm: Breadth and Pragmatism vs. Depth and Originality.**    The ACM-ICPC is fundamentally a test of **problem-solving breadth and rapid implementation**. A typical contest

features a large number of problems (8–13) to be solved within a tight five-hour window. This structure rewards a broad, practical knowledge of standard algorithms and data structures, and the ability to quickly recognize a problem pattern and apply a known, reliable solution. This directly parallels the day-to-day reality of a software engineer, who must efficiently address a wide variety of tasks, from implementing new features to fixing bugs, by applying the right tool for the job.

In contrast, the IOI is a test of **algorithmic depth and creative invention**. With only a few (typically three) highly complex problems per day, it challenges participants to devise novel or highly optimized algorithms, often pushing the boundaries of known techniques. This format is more akin to academic research or work in a specialized R&D lab. For an agent intended to serve as a general-purpose coding assistant, the broad-based, pragmatic skill set measured by the ICPC is a more relevant benchmark.

**Scoring Philosophy: The Imperative of Zero-Bug Correctness.** A defining feature of the ACM-ICPC is its **all-or-nothing scoring system**. A submission earns credit if and only if it passes every single hidden test case. A solution that fails on a single edge case is equivalent to one that fails completely. This binary outcome brutally enforces the concept of **robustness and correctness**, which is the bedrock of reliable software engineering. In a production environment, code with a "99% pass rate" is simply buggy code, and a single critical failure can have catastrophic consequences. The ICPC format, therefore, directly measures an agent's ability to produce *zero-bug* solutions.

Furthermore, the ICPC's penalty system—which adds a fixed time penalty for each incorrect submission on a problem that is eventually solved—explicitly disincentivizes a careless trial-and-error approach. It rewards careful planning, local testing, and a deep consideration of edge cases before submission, all of which are hallmarks of a disciplined engineering process.

The IOI, conversely, employs a **partial-credit system**, awarding points based on the number of test cases a solution correctly handles. This is excellent for measuring incremental progress and rewarding clever heuristics that solve a subset of the problem. However, it does not instill the same absolute imperative for correctness that the ICPC format does. For an agent destined for real-world deployment, the ICPC's unforgiving standard of correctness is a far more meaningful and critical measure of its reliability.

**Alignment with Engineering Values: Efficiency and Delivery over Theoretical Optimality.** The intense time pressure and large problem set of the ACM-ICPC naturally encourage competitors to find the **simplest, most direct path to a correct solution**. The goal is not necessarily to write the most theoretically optimal or elegant code, but to write *correct and efficient enough* code to pass within the given constraints, and to do so quickly. This mindset perfectly mirrors the agile, delivery-focused nature of modern software development, where delivering a working, maintainable feature on schedule is paramount, and premature optimization is a well-known anti-pattern.

The IOI's focus on a small number of extremely difficult problems, in contrast, incentivizes the pursuit of **theoretical optimality**. The challenge often lies in shaving off logarithmic factors in complexity or designing a complex algorithm that precisely meets stringent time and memory limits. While an exceptional display of algorithmic prowess, this is often a form of over-engineering in a typical software development context. An agent that rapidly delivers a correct $O(N \log N)$ solution is often more valuable than one that spends immense resources to discover a complex $O(N)$ solution, especially when the former is sufficient for the task at hand.

In summary, by adopting the ACM-ICPC format, we are explicitly choosing to evaluate coding agents against a set of criteria that prioritize the core values of software engineering: broad applicability, rigorous correctness, and the efficient delivery of robust solutions under constraints. This makes USACOArena not just a test of algorithmic knowledge, but a direct measure of an agent's potential as a practical and reliable engineering tool.

## D    PROBLEM CORPUS AND DIFFICULTY BASELINING

**Corpus Philosophy**    The problem corpus for USACOArena is not a fixed, static set. It is a living collection that mirrors the official USACO contest schedule. For each of the four contests in a USACO season, we adopt the official 12-problem set—three problems each for Bronze, Silver, Gold,

and Platinum levels—as the basis for a distinct USACOArena competition. This approach ensures that our benchmark stays current with the evolving difficulty and style of competitive programming problems and avoids any potential bias from manual problem curation.

**Difficulty Baselining Study** To provide an empirical baseline of difficulty for the novel problems in the 2024-2025 season, we conduct an analysis using a top-tier agent, Gemini-2.5-pro. Each of the 48 problems is run with ample resources to assess its inherent solvability by a state-of-the-art model. The results of this study, presented in Table 3, are not used to select or filter problems, but rather to provide a grounded reference for analyzing agent performance in the main experiments. For example, this data helps us understand when an agent fails on a problem that is known to be solvable, indicating a potential strategic failure rather than a fundamental capability gap.

Table 3: Difficulty baselining results for the 48 problems of the USACO 2024-2025 season, using Gemini-2.5-pro. This data provides a grounded reference for expected problem difficulty in our main experiments.

| Problem ID | Level | Result | Test Cases | Cons. Credit | LLM Calls | Sub. |
|---|---|---|---|---|---|---|
| 1526 | platinum | WA | 1/20 | 10167024 | 48 | 37 |
| 1525 | platinum | WA | 1/15 | 10306989 | 45 | 36 |
| 1524 | platinum | TLE | 1/16 | 10018138 | 63 | 49 |
| 1523 | gold | TLE | 1/20 | 10151386 | 57 | 47 |
| 1522 | gold | AC | 15/15 | 1446214 | 6 | 1 |
| 1521 | gold | AC | 25/25 | 2818862 | 18 | 8 |
| 1520 | silver | AC | 17/17 | 2936761 | 17 | 14 |
| 1519 | silver | AC | 12/12 | 7132837 | 36 | 27 |
| 1518 | silver | WA | 0/18 | 10131801 | 55 | 45 |
| 1517 | bronze | AC | 11/11 | 510085 | 3 | 1 |
| 1516 | bronze | AC | 11/11 | 204672 | 2 | 1 |
| 1515 | bronze | AC | 12/12 | 137433 | 3 | 1 |
| 1502 | platinum | WA | 0/20 | 10164077 | 45 | 21 |
| 1501 | platinum | WA | 1/19 | 10035673 | 55 | 42 |
| 1500 | platinum | WA | 1/13 | 10065047 | 54 | 38 |
| 1499 | gold | AC | 18/18 | 252597 | 2 | 1 |
| 1498 | gold | TLE | 1/20 | 10012120 | 52 | 45 |
| 1497 | gold | AC | 21/21 | 2939562 | 10 | 3 |
| 1496 | silver | AC | 12/12 | 583964 | 4 | 2 |
| 1495 | silver | AC | 18/18 | 628954 | 4 | 3 |
| 1494 | silver | TLE | 1/18 | 10099233 | 53 | 35 |
| 1493 | bronze | AC | 13/13 | 214167 | 2 | 1 |
| 1492 | bronze | AC | 11/11 | 201263 | 2 | 1 |
| 1491 | bronze | AC | 16/16 | 104275 | 2 | 1 |
| 1478 | platinum | WA | 2/19 | 10175416 | 45 | 38 |
| 1477 | platinum | TLE | 1/14 | 10031850 | 46 | 28 |
| 1476 | platinum | AC | 23 | 735288 | 4 | 3 |
| 1475 | gold | WA | 4/18 | 10129632 | 50 | 39 |
| 1474 | gold | AC | 23/23 | 2377487 | 12 | 10 |
| 1473 | gold | AC | 16/16 | 1723281 | 8 | 2 |
| 1472 | silver | AC | 15/15 | 1368787 | 10 | 8 |
| 1471 | silver | AC | 16/16 | 411089 | 3 | 1 |
| 1470 | silver | AC | 23/23 | 420372 | 3 | 1 |
| 1469 | bronze | AC | 13/13 | 153411 | 2 | 1 |
| 1468 | bronze | AC | 11/11 | 420033 | 4 | 3 |
| 1467 | bronze | AC | 12/12 | 472262 | 4 | 2 |
| 1454 | platinum | WA | 1/20 | 10146203 | 70 | 26 |
| 1453 | platinum | TLE | 1/18 | 10067438 | 84 | 19 |
| 1452 | platinum | AC | 15/15 | 571806 | 3 | 2 |

Table 3: – continued from previous page

| Problem ID | Level | Result | Test Cases | Cons. Credit | LLM Calls | Submissions |
|---|---|---|---|---|---|---|
| 1451 | gold | AC | 16/16 | 1151155 | 6 | 4 |
| 1450 | gold | AC | 23/23 | 4026173 | 26 | 13 |
| 1449 | gold | AC | 20/20 | 1119907 | 6 | 5 |
| 1448 | silver | AC | 13/13 | 2734345 | 11 | 2 |
| 1447 | silver | AC | 11/11 | 2035808 | 8 | 1 |
| 1446 | silver | AC | 11/11 | 1649536 | 8 | 5 |
| 1445 | bronze | AC | 13/13 | 373173 | 3 | 2 |
| 1444 | bronze | AC | 16/16 | 102214 | 3 | 1 |
| 1443 | bronze | AC | 13/13 | 311834 | 2 | 1 |
| 1430 | platinum | TLE | 4/24 | 10106758 | 55 | 45 |
| 1429 | platinum | TLE | 1/22 | 10170320 | 56 | 41 |
| 1428 | platinum | AC | 25/24 | 7526267 | 40 | 32 |
| 1427 | gold | AC | 20/20 | 1213337 | 7 | 4 |
| 1426 | gold | AC | 20/20 | 479578 | 3 | 1 |
| 1425 | gold | AC | 23/23 | 374396 | 3 | 2 |
| 1424 | silver | AC | 16/16 | 528027 | 3 | 1 |
| 1423 | silver | AC | 15/15 | 951701 | 5 | 2 |
| 1422 | silver | WA | 1/21 | 10032808 | 74 | 44 |
| 1421 | bronze | TLE | 2/11 | 10098563 | 66 | 51 |
| 1420 | bronze | AC | 11/11 | 173543 | 2 | 1 |
| 1419 | bronze | AC | 26/26 | 221140 | 2 | 1 |

# E  PARTICIPANT QUALIFICATION RESULTS

To ensure that the agents evaluated in our main experimentspossess a baseline of functional competency, we implemented a qualification stage for each of the four USACO contests. The core requirement was for an agent to successfully solve the single easiest Bronze-level problem from the respective contest set. Only agents that achieved an 'Accepted' (AC) status on this prerequisite task were included in the full, multi-problem competitive runs analyzed in the main paper.

The following tables (Table 4 through Table 7) provide the detailed performance results for this qualification task across the four contests. These results not only justify our selection of participants for the main analysis but also offer a preliminary glimpse into the vast performance disparities among the models. Even on these relatively simple entry-level problems, we observe significant variance in resource consumption (Consumed Credit) and efficiency (Submissions), foreshadowing the more complex strategic differences analyzed in the main text.

Table 4: Representative qualification results for the USACO 2024 December Contest. Agents are required to solve the easiest Bronze problem 1445. Cons. Credit means Consumed Credit.

| Model | Result | Cons. Credit | LLM Calls | Submissions | Qualified |
|---|---|---|---|---|---|
| Gemini-2.5-Pro | AC | 373,173 | 3 | 2 | Yes |
| GPT-5-Codex | AC | 26,986 | 3 | 1 | Yes |
| Claude-4-Sonnet | AC | 57,405 | 3 | 2 | Yes |
| DeepSeek-V3.1 | AC | 69,514 | 19 | 7 | Yes |
| DeepSeek-V3 | AC | 536,880 | 130 | 47 | Yes |
| Kimi-K2-0905 | AC | 449,065 | 35 | 17 | Yes |
| Qwen3-235B | AC | 495,102 | 49 | 7 | Yes |
| GLM-4.5 | AC | 7,725 | 2 | 1 | Yes |

Table 5: Representative qualification results for the USACO 2025 January Contest. To qualify, agents were required to solve at least one of the three available Bronze problems (1467, 1468, 1469).

| Model | Result | Cons. Credit | LLM Calls | Submissions | Qualified |
|---|---|---|---|---|---|
| Gemini-2.5-Pro | AC | 472,262 | 4 | 2 | Yes |
| GPT-5-Codex | AC | 177,127 | 9 | 1 | Yes |
| | WA | 10,035,455 | 168 | 90 | |
| Claude-4-Sonnet | TLE | 10,037,109 | 508 | 232 | No |
| | TLE | 10,013,656 | 227 | 127 | |
| DeepSeek-V3.1 | AC | 497,429 | 121 | 46 | Yes |
| Kimi-K2-0905 | AC | 2,605,011 | 196 | 89 | Yes |
| Qwen3-235B | AC | 99,574 | 10 | 5 | Yes |
| GLM-4.5 | AC | 245,624 | 29 | 5 | Yes |

Table 6: Representative qualification results for the USACO 2025 February Contest. Agents are required to solve the easiest Bronze problem 1491. Cons. Credit means Consumed Credit.

| Model | Result | Cons. Credit | LLM Calls | Submissions | Qualified |
|---|---|---|---|---|---|
| Gemini-2.5-Pro | AC | 104,275 | 2 | 1 | Yes |
| GPT-5-Codex | AC | 19,739 | 3 | 1 | Yes |
| Claude-4-Sonnet | AC | 57,486 | 3 | 2 | Yes |
| DeepSeek-V3.1 | AC | 14,048 | 5 | 1 | Yes |
| DeepSeek-V3 | AC | 6,918 | 4 | 1 | Yes |
| Kimi-K2-0905 | AC | 213,684 | 18 | 9 | Yes |
| Qwen3-235B | AC | 122,844 | 12 | 3 | Yes |
| GLM-4.5 | AC | 7,474 | 2 | 1 | Yes |

Table 7: Representative qualification results for the USACO 2025 US Open Contest. Agents are required to solve the easiest Bronze problem 1515. Cons. Credit means Consumed Credit.

| Model | Result | Cons. Credit | LLM Calls | Submissions | Qualified |
|---|---|---|---|---|---|
| Gemini-2.5-Pro | AC | 137433 | 3 | 1 | Yes |
| GPT-5-Codex | AC | 37583 | 4 | 1 | Yes |
| Claude-4-sonnet | AC | 813746 | 20 | 9 | Yes |
| DeepSeek-V3.1 | AC | 1153921 | 257 | 47 | Yes |
| DeepSeek-V3 | AC | 239910 | 55 | 17 | Yes |
| Kimi-K2 | AC | 1366068 | 95 | 23 | Yes |
| Qwen3-235B | AC | 308213 | 24 | 4 | Yes |
| GLM-4.5 | AC | 245624 | 29 | 5 | Yes |

## F    DETAILED RESULTS OF MAIN COMPETITION

This section provides the detailed results from our main experiment, where each qualified agent competed in the full 12-problem USACOArena contest. To account for the stochastic nature of agent performance and potential variations in LLM API responses, each agent completed the competition five times. The results presented in the main paper are the average of these five runs.

Table 8 presents the aggregated performance metrics for each agent, including the average rank, score, and consumed credit, along with their standard deviations. We also provide a breakdown of the average credit consumption across the main categories—LLM inference, hints, and penalties—to offer deeper insight into each agent's prevailing strategy. The agents are sorted by their final average rank, determined first by average rank and then by average score and consumed credit.

The raw, run-by-run data for each agent, including detailed action logs and final scores for each of the five trials, are available in the supplementary material for full reproducibility.

Table 8: Aggregated results from the main experiment, averaged over 5 runs across four contests. The data shows each agent's final rank, score, and credit consumption, reflecting their strategic priorities. Values are presented as mean ± standard deviation.

| Model | Avg. Rank | Avg. Score | Avg. Consumed Credit | Inference Credit | Hint Credit | Penalty Credit |
|---|---|---|---|---|---|---|
| Gemini-2.5-pro | 1.30 ± 0.47 | 14.00 ± 3.88 | 13,762,787 ± 4.3M | 13.76M ± 4.3M | 2.5K ± 2.5K | 4.1K ± 1.9K |
| GPT-5-Codex | 1.70 ± 0.47 | 9.39 ± 7.59 | 4,707,464 ± 3.1M | 4.71M ± 3.1M | 1.1K ± 2.0K | 0.3K ± 0.3K |
| Qwen3-235b | 4.00 ± 1.59 | 1.61 ± 0.92 | 11,732,391 ± 6.9M | 11.17M ± 6.5M | 560.2K ± 375.7K | 3.6K ± 2.6K |
| GLM-4.5 | 4.35 ± 1.57 | 2.33 ± 2.28 | 7,249,215 ± 4.0M | 7.06M ± 3.9M | 161.7K ± 93.2K | 22.7K ± 16.8K |
| DeepSeek-V3 | 5.70 ± 1.13 | 0.11 ± 0.32 | 194,050 ± 0.2M | 0.17M ± 0.2M | 19.8K ± 24.9K | 1.2K ± 1.3K |
| DeepSeek-V3.1 | 6.00 ± 1.30 | 0.06 ± 0.24 | 253,013 ± 0.4M | 0.23M ± 0.4M | 21.9K ± 32.7K | 1.4K ± 2.7K |
| Kimi-K2-0905 | 6.00 ± 1.45 | 0.72 ± 1.18 | 1,337,561 ± 2.0M | 1.32M ± 2.0M | 10.0K ± 17.2K | 3.0K ± 4.2K |
| Claude-4-sonnet | 6.95 ± 1.36 | 0.39 ± 0.61 | 1,285,766 ± 0.8M | 1.28M ± 0.8M | 1.4K ± 1.7K | 1.1K ± 0.6K |

Table 9: Performance and Strategy Comparison within the GPT-5 Agent Family. The value in parenthesis indicates the head-to-head win rate for each agent within its matchup.

| Agent | Win Rate | Avg. Score | Avg. Credit Consumed | Attempted Problems | Submission Precision (%) |
|---|---|---|---|---|---|
| *Experiment 1: Generalist vs. Specialist* | | | | | |
| GPT-5 (Base) | **100%** | **20.3** | 14M | 26 | 12.9% |
| GPT-5-Codex | 0% | 8.0 | 8M | 16 | 68.2% |
| *Experiment 2: Specialist vs. Agentic Framework* | | | | | |
| GPT-5-Codex | 0% | 5 | 4M | 16 | 44.4% |
| Codex-CLI | **100%** | **9.5** | 15M | 21 | 52.4% |

The aggregated results highlight key strategic differences. For example, while GPT-5 and Gemini-2.5-pro are the clear top performers, GPT-5 consistently consumes less credit across all categories, indicating a more efficient problem-solving process. The credit breakdown also reveals that lower-tier models often accumulate significant penalty credit without a corresponding increase in score, suggesting a tendency towards inefficient trial-and-error strategies.

# G PROBING AGENT ARCHITECTURE: A CASE STUDY ON THE GPT-5 FAMILY

To isolate the impact of different design choices, we conduct controlled "civil war" experiments within the GPT-5 family. We run two head-to-head matchups: the base GPT-5 against its code-specialized variant, GPT-5-Codex; and GPT-5-Codex against Codex-CLI, a version augmented with an agentic framework. Each matchup is run three times on the challenging US Open contest problem set, with the averaged results reported in Table 9. In our analysis, submission precision is defined as the ratio of correct submissions to the total number of attempts.

Our experiments reveal key trade-offs in agent development. In the first matchup, the specialized GPT-5-Codex adopts a far more cautious approach than its base model. It frequently uses the TEST action to ensure high submission precision, but its reluctance to attempt uncertain problems limited its overall score. This suggests that code-specialization enhances reliability, potentially at the cost of problem-solving initiative. The second matchup shows that the agentic framework on Codex-CLI significantly improves performance, achieving a much higher score and win rate while maintaining the same high precision. This demonstrates that a well-designed architecture can directly boost a model's performance, though this may come with higher resource consumption.

# H LARGE LANGUAGE MODEL DETAILS

Our evaluation leverages a diverse suite of Large Language Models (LLMs) to ensure a comprehensive analysis of agent capabilities within the USACOArena. Table 10 provides a detailed breakdown of the models employed in our study, including their provider and associated costs. All pricing data,

specified in U.S. dollars per million input and output tokens respectively, was retrieved from Artificial Analysis[2] in September 2025. This selection represents a cross-section of the contemporary LLM landscape, encompassing models with varied architectures, parameter scales, and economic costs, thereby facilitating a robust and multifaceted analysis of agent performance.

Table 10: Specifications of Large Language Models used in our evaluation. Costs are denoted in USD per million tokens.

| Provider | Model | Input Cost | Output Cost |
|---|---|---|---|
| OpenAI | GPT-5-2025-08-07 | $1.25 | $10.00 |
| | GPT-5-Codex | $1.25 | $10.00 |
| Google | Gemini 2.5 Pro | $1.25 | $10.00 |
| Anthropic | Claude-Sonnet-4-20250514 | $3.00 | $15.00 |
| DeepSeek | DeepSeek-v3 | $0.27 | $1.10 |
| | DeepSeek-v3.1 | $0.27 | $1.10 |
| Alibaba Cloud | Qwen3-235B-A22B-Instruct-2507 | $0.70 | $2.80 |
| Moonshot AI | Kimi-K2-0905 | $1.00 | $2.75 |
| Zhipu AI | GLM-4.5 | $0.59 | $2.19 |

---

[2]https://artificialanalysis.ai

# I USACOARENA HYPERPARAMETERS

The main experiments conducted in this study utilize a standardized default configuration for the USACOArena environment. This configuration, which is highly customizable to facilitate diverse research questions, is detailed in Table 11.

Table 11: USACOArena Competition Configuration Parameters

| Parameter | Description | Default Value |
|---|---|---|
| *Basic Setup* | | |
| max_credits_per_participant | Maximum credits per participant | 20,000,000 |
| *Scoring System* | | |
| bronze_score | Points for Bronze problems | 1 |
| silver_score | Points for Silver problems | 2 |
| gold_score | Points for Gold problems | 5 |
| platinum_score | Points for Platinum problems | 10 |
| *LLM Inference* | | |
| agent_temperature | Model generation temperature | 0.7 |
| *Hint Request Costs* | | |
| level_0_hint | Strategy hints cost | 500 |
| level_1_hint | Textbook knowledge cost | 1,000 |
| level_2_hint | Knowledge-specific content cost | 1,000 |
| level_3_hint | Similar problems cost | 1,500 |
| level_4_hint | Example problems cost | 1,500 |
| *Test Code Costs* | | |
| test_code | Base cost per test request | 10 |
| *Penalty System* | | |
| WA_penalty | Penalty for Wrong Answer | 100 |
| RE_penalty | Penalty for Runtime Error | 100 |
| CE_penalty | Penalty for Compile Error | 100 |
| TLE_penalty | Penalty for Time Limit Exceeded | 100 |
| MLE_penalty | Penalty for Memory Limit Exceeded | 100 |
| *Problem Distribution* | | |
| total_problems | Total number of problems | 12 |
| bronze_problems | Bronze difficulty count | 3 |
| silver_problems | Silver difficulty count | 3 |
| gold_problems | Gold difficulty count | 3 |
| platinum_problems | Platinum difficulty count | 3 |

# J PROMPT FOR USACOARENA EVALUATION

The following box details the complete prompt structure provided to agents in the USACOArena competition. The prompt is designed as a purely objective specification of the environment. It comprehensively delineates the foundational components of the competition: the governing rules, the format for communicating game state, the complete set of available actions, and the structure of action results. Crucially, the prompt deliberately refrains from offering any strategic guidance or heuristics. This ensures that all observed strategies are properties of the agent's autonomous decision-making process, rather than a reflection of guidance embedded in the instructions

---

**An Example Prompt for Evaluating Agentic LLMs in USACOArena**

SYSTEM PROMPT

You are a competitive programming agent participating in a coding competition. You will receive the current state of the competition and results of your previous actions. Your goal is to solve as many problems as possible (achieve 'Accepted' status).

Your final ranking is determined first by your total score. The score is a weighted sum of the problems you solve, with harder problems (e.g., Platinum) being worth more than easier ones (e.g., Bronze). If scores are tied, the agent with the lower total of (*actual consumed credit + penalties*) ranks higher.

You start with a limited credit budget, and many actions consume credit. **You will be terminated from the competition when your actual consumed credit reaches the limit.**

Credit is consumed in three main ways:

1. **LLM Inference:** Generating responses, which consumes credit based on the number of tokens you use.

2. **Purchasing Hints:** Using hints to help solve problems.

3. **Testing Code:** Running your code against test cases before final submission.

**IMPORTANT:**

- Penalties from wrong submissions affect your ranking tie-breaker but do **NOT** count toward termination.

- In this competition, solving problems is much more important than minimizing the consumed credit. So you should try your best to solve as many problems as possible.

Please respond with a JSON object containing 'action' and 'parameters' fields.

- - - - - - - - - - - - - - - - - - - - - - - - - - - - - - - - - -

USER PROMPT

COMPETITION RULES

- **Credit System:**
  - Each participant starts with a total of **20,000,000** credit limit.
  - Credit is consumed by three main sources: LLM Inference, Purchasing Hints, and Testing Code.
  - Your participation ends when your **actual consumed credit** reaches the limit.

- **Scoring Rules:**
  - Your **Final Score** is the sum of points from all problems you solve completely.
  - No partial credit is awarded.
  - Points are weighted by difficulty: Bronze (1), Silver (2), Gold (5), Platinum (10).

- **Penalties:** A penalty of 100 points is incurred for CE, MLE, RE, TLE, and WA submissions.

- **Ranking and Tie-Breaking:** Rank is determined by Final Score. Ties are broken by the lower (Actual Consumed Credit + Penalties).

- **Programming Languages:** C++17, Java, and Python3 are available.

YOUR STATUS

- **Name:** `<agent_name>`
- **Consumed Credit:** `<consumed_credit>`
- **Solved Problems:** `<solved_list>`
- **Current Score:** `<score>`
- **Penalty:** `<penalty>`

AVAILABLE PROBLEMS

- `<problem_id_1>`
- `<problem_id_2>`
- `...`

CURRENT RANKINGS

```
1. <Agent 1>: Score <S1>, Credit+Penalty: <C1> [ACTIVE]
2. <Agent 2>: Score <S2>, Credit+Penalty: <C2> [TERMINATED]
...
```

AVAILABLE ACTIONS

1. **VIEW_PROBLEM:** View problem details.
2. **GET_HINT:** Get a hint for a problem (consumes credit). Levels 0-4 are available.
3. **SUBMIT_SOLUTION:** Submit a solution.
4. **TEST_CODE:** Test code with custom test cases (consumes credit).
5. **TERMINATE:** End participation.

RESPONSE FORMAT

Please respond using the following JSON format:

```
{
  "action": "<action_name>",
  "parameters": {
    // Fill in parameters according to the action type
  }
}
```

## K    THE FIVE-TIERED HINT SYSTEM IN USACOARENA

Empirically, while current top-tier agents rarely utilize this hint infrastructure, its design is far from over-engineered; rather, it serves as a critical **Capability Probe**. The observation that agents consistently fail to purchase hints even when hopelessly stuck in a loop reveals a fundamental *Metacognitive Deficit*. Current models lack the self-awareness to evaluate their own progress and recognize *when* they require external theoretical intervention. The five-tiered system is an essential instrument for measuring the exact moment future architectures bridge this metacognitive gap.

To rigorously evaluate an agent's ability to make strategic decisions under resource constraints, we engineered a sophisticated five-tiered hint system within USACOArena. This system is not merely a help feature; it functions as an economic model where information is a commodity with varying costs and utilities. Agents must perform a cost-benefit analysis to decide if, when, and what type of hint to purchase. This design allows us to observe and quantify an agent's resource management and problem-solving strategies.

**Level 0: Strategic Guidance (500 Credit)**    This foundational hint provides high-level, static information about competitive programming.

- **Function:** It delivers a pre-compiled document containing the core philosophy of competitive programming, a comprehensive debugging checklist, and general contest strategies (e.g., time management). The contents are derived from USACO Guide[3].
- **Mechanism:** The system retrieves the full content from a static JSON file (`/dataset/corpuses/USACO_strategy.json`). The API call is parameter-free (`{"hint_level": 0}`).
- **Strategic Purpose:** This low-cost hint is intended for the early stages of a competition, allowing an agent to establish a baseline understanding of the meta-game without spending significant resources.

**Level 1: Problem-Specific Textbook Content (1,000 Credit)**    This hint offers theoretical knowledge directly relevant to a specific problem.

- **Function:** It provides a concise, relevant excerpt from a competitive programming textbook that explains the theoretical concepts or algorithms needed for a given problem.
- **Mechanism:** Upon receiving a `problem_id`, the system automatically extracts key algorithmic and data structure terms from the problem description. It then employs a BM25 search algorithm to find the most relevant section in a 2.8MB textbook corpus which is de-

---

[3]`https://usaco.guide`

rived from Algorithms for Competitive Programming[4]. The top result is returned, truncated to 1,000 characters.

- **Strategic Purpose:** This is for agents that can identify a knowledge gap related to a problem but do not know the name of the required algorithm. It tests the agent's ability to recognize when it needs theoretical grounding.

**Level 2: Knowledge-Targeted Textbook Content (1,000 Credit)**   Similar to Level 1, this hint also retrieves textbook content, but with a key difference in agent interaction.

- **Function:** It provides a detailed explanation of a specific algorithm or data structure explicitly named by the agent.
- **Mechanism:** Instead of a `problem_id`, the agent must provide a `hint_knowledge` keyword (e.g., "segment tree"). The system uses this keyword directly in its BM25 search against the same textbook corpus.
- **Strategic Purpose:** This hint is for a more advanced scenario where an agent correctly identifies the required algorithm by name but needs to learn its implementation details. It tests an agent's self-awareness of its specific knowledge deficits.

**Level 3: Similar Problem Retrieval (1,500 Credit)**   This high-cost hint provides a concrete, solved example of a similar problem.

- **Function:** It returns a full problem description, along with a complete, vetted solution and explanation, for a problem that is semantically similar to the one the agent is currently working on.
- **Mechanism:** The system uses the current `problem_id`'s text (description and samples) as a query for a BM25 search against the entire USACO problem library derived from USACO Guide[5], excluding problems from the current competition. The most similar problem is returned.
- **Strategic Purpose:** This is a powerful tool for agents that are completely stuck on the problem-solving approach. Its high cost forces the agent to consider whether viewing a direct analogy is worth the significant credit expenditure.

**Level 4: Curated Example Problems (1,500 Credit)**   This is the most targeted hint, designed to provide practice on a specific topic at a specific difficulty.

- **Function:** It retrieves a complete example problem (description, solution, complexity analysis) that matches both a user-specified difficulty level and a knowledge keyword.
- **Mechanism:** The system filters the entire USACO problem library based on both `problem_difficulty` (e.g., "Bronze") and `hint_knowledge` (e.g., "complete search") tags provided by the agent.
- **Strategic Purpose:** This hint allows an agent to request a targeted exercise, simulating a human's process of looking for practice problems. It tests the agent's ability to formulate a precise learning objective.

---

[4]`https://cp-algorithms.com`
[5]`https://usaco.guide`

