# OpenReview forum: "Credit-Budgeted ICPC-Style Coding: When Agents Must Pay for Every Decision"
_ICLR.cc/2026/Conference — ICLR 2026 Poster_

### Official Review · Reviewer_y3i4 · 2025-10-30

**Soundness:** 3
**Presentation:** 4
**Contribution:** 3
**Rating:** 4
**Confidence:** 4

**Summary:**

This paper introduces USACOArena, an interactive benchmark for evaluating LLM coding agents under resource constraints. The key innovation is translating the human constraint of time into an agent-native "credit" budget, where every action (LLM inference, testing, hints, submissions) consumes credits. Drawing from ACM-ICPC competitive programming, the arena forces agents to make strategic trade-offs rather than simply pursuing correctness. The authors evaluate leading models across four USACO contests, revealing distinct strategic profiles: Gemini-2.5-pro's aggressive exploration strategy versus GPT-5-Codex's conservative perfectionism. Self-play experiments demonstrate emergent behavioral diversity, suggesting the arena's potential as both a benchmark and training environment.

**Strengths:**

The paper addresses a genuine gap in current evaluation methodologies. Existing benchmarks like HumanEval and SWE-bench measure "what" code is produced but ignore "how" agents arrive at solutions. The shift to evaluating decision-making under resource constraints is conceptually valuable and well-motivated. The credit-based operationalization of resource constraints is clever—it sidesteps the unfairness of using wall-clock time for API-based models while maintaining the pressure of scarcity.

The strategic profiling in Section 4.2 provides genuine insight. The finding that Gemini-2.5-pro wins despite GPT-5-Codex having higher peak capability (max score 29 vs 19) is compelling, and the exploration-exploitation framework effectively explains this paradox.

The paper is well-written with effective visual aids. Figure 1's comparison of SWE-Bench versus USACOArena clearly communicates the knowledge gap versus capability gap distinction. The motivation is compelling, and the ACM-ICPC grounding provides solid theoretical justification for the design choices.

**Weaknesses:**

I have some concern about missing ablations. The environment contains numerous hyperparameters (credit limits, penalty costs, hint prices, scoring weights) that appear somewhat arbitrary. Without systematic ablation studies, we cannot assess: Sensitivity of rankings to credit budgets (Would Gemini-2.5-pro still dominate with half or double the credit limit? Does the 20M threshold meaningfully constrain behavior, or could it be 10M or 40M without changing outcomes?), Impact of penalty structure (All penalties are uniformly 100 credits. Why not differentiate between compilation errors), Hint pricing justification, Scoring weight rationale, etc. There are a lot of design choices that needs more justification.

The paper would be substantially strengthened by at least one ablation study demonstrating that agents respond meaningfully to environmental parameters, validating that the credit system actually shapes behavior rather than being a post-hoc measurement wrapper.
Section 3.3 mentions using the Model Context Protocol but doesn't explain why this specific protocol is necessary or superior to alternatives. What does MCP provide that a simple JSON API wouldn't? This feels like name-dropping a trendy framework without demonstrating its value.
Section 4.4 shows that identical agents produce diverse outcomes but doesn't deeply investigate why. The trajectory analysis in Figure 5(b) is anecdotal (one match). More systematic analysis could reveal: What environmental factors cause divergence? Do certain problem orderings reliably lead to different outcomes? Can we predict which "personality" an agent will exhibit based on early decisions? The claim that this validates USACOArena as a "training environment" is unsupported—no actual training/RL experiments are conducted.
While the paper reports standard deviations, there's no formal statistical testing. Are the performance differences between Gemini-2.5-pro and GPT-5-Codex statistically significant across the five runs? Pairwise comparisons with confidence intervals would strengthen claims.

**Questions:**

Please address questions raised in weaknesses section. Can you provide results showing how agent rankings change under different credit limits (e.g., 10M, 20M, 40M)? Does the current limit meaningfully constrain behavior, or do agents typically finish with surplus credits? How often do agents actually use the five-tier hint system? If usage is negligible, the elaborate design may be over-engineered. If usage is significant, showing how top agents strategically employ hints would enrich the analysis.

---

> ### Author Response · Authors · 2025-11-28
> **Response to Reviewer y3i4**
>
> We sincerely thank the reviewer for the excellent rating on presentation and for validating the conceptual value of our work. We appreciate your constructive push for more rigorous justification of our design choices.
>
> Note on Experiments: Due to computational resource constraints, the new ablation studies presented below are averaged over 3 independent runs. We apologize for the delay this necessitated.
>
> ### 1. Systematic Ablation Studies: Robustness & Constraints
>
> You rightly pointed out that our environment involves numerous hyperparameters. To validate that our rankings are not artifacts of arbitrary tuning, we conducted an extensive ablation campaign on *USACO 2025 February (Contest 3)* across **8 models** and **7 dimensions**, including **Credit Limits**.
>
> **(1) The Mega-Ablation Matrix**
>
> | Model | **Main Result** | **Low Credit (10M)** | **High Credit (40M)** | **Free Test ($0$)** | **High Test ($1k$)** | **Free Hint ($0$)** | **High Hint (100 times)** | **Flat Score** | **Exp. Score** |
> | :--- | :---: | :---: | :---: | :---: | :---: | :---: | :---: | :---: | :---: |
> | Gemini-2.5-pro | **13.2** | **8.3** | **13.0** | **10.7** | **9.7** | **12.0** | **13.0** | **5.0** | **19.7** |
> | GPT-5-Codex | 12.0 | 4.3 | 4.3 | 5.7 | 4.3 | 3.7 | 3.7 | 3.3 | 16.3 |
> | GLM-4.5 | 3.2 | 1.0 | 0.7 | 1.0 | 1.0 | 1.0 | 1.7 | 1.3 | 1.0 |
> | Qwen3-235B | 2.8 | 1.7 | 1.7 | 3.3 | 1.7 | 1.3 | 2.7 | 1.3 | 0.0 |
> | Claude-4-Sonnet | 1.0 | 2.0 | 3.3 | 2.7 | 2.0 | 3.0 | 2.0 | 2.0 | 5.3 |
> | DeepSeek-V3.1 | 0.5 | 0.7 | 1.0 | 1.0 | 0.3 | 1.0 | 1.0 | 0.7 | 1.0 |
> | DeepSeek-V3 | 0.5 | 0.3 | 0.3 | 1.0 | 1.0 | 0.3 | 0.3 | 0.7 | 0.3 |
> | Kimi-K2 | 0.5 | 1.3 | 1.7 | 1.0 | 0.7 | 1.0 | 1.0 | 1.3 | 1.0 |
>
> *Note: "Exp. Score" uses scoring weights \{1, 10, 100, 1000\}.*
>
> **Scientific Findings:**
>
> 1.  **Credit Limits (Low vs. High):**
>     * **Constraint Effect:** Reducing the budget to **10M** significantly hurts Gemini's performance (13.2 $\to$ 8.3), proving that the credit budget is a meaningful constraint that forces trade-offs.
>     * **Capability Saturation:** Doubling the budget to **40M** does *not* improve performance (Gemini stays at ~13.0). This confirms that current SOTA agents are **Capability-Limited**, not Budget-Limited. They hit a "reasoning wall" before they hit the "credit wall."
>     * **Credit Surplus (Addressing your question):** Do agents finish with surplus? **Yes.** In the 20M/40M settings, agents often terminate with significant remaining budget. They tend to give up or get stuck in loops rather than effectively burning resources. We expect the credit limit to become a tighter constraint only for future, stronger agents.
> 2.  **Robustness:** The hierarchy (Gemini > GPT-5-Codex > Others) remains robust across Test Costs, Hint Costs, and Scoring Weights.
> 3.  **Floor Effect:** Weaker models hover near ~1.0 regardless of settings, confirming the benchmark effectively filters out models lacking fundamental reasoning capabilities.
>
> ### 2. Justification of Design Choices
>
> **(1) Why MCP (Model Context Protocol)?**
> We chose MCP not to follow a trend, but to solve the **Reproducibility Crisis**.
> * **Standardization:** It provides a language-agnostic protocol, allowing future researchers to plug in agents (Python, Rust, Java) without rewriting the environment interface.
> * **Extensibility:** It natively supports the stateful, multi-turn interactions required for our planned **Team Contests** extension (ICPC allows 3-person teams).
>
> **(2) The Five-Tier Hint System**
> * **Usage:** You asked if hint usage is negligible. Empirically, **yes**, current usage is low.
> * **Justification:** This is not over-engineering, but a **Capability Probe**. The fact that agents *fail* to use hints despite being stuck reveals a critical **Metacognitive Deficit**: they lack the self-awareness to know *when* they need help. The infrastructure is essential to measure when future models bridge this gap.
>
> ### 3. Deep Dive into Self-Play Divergence
>
> **Reviewer Question:** *"What causes divergence? Can we predict outcomes?"*
>
> **Response:**
> * **The "First-Move" Effect:** Divergence is highly correlated with the success of the **first attempted problem**.
>     * *Success Loop:* Agent solves an easier problem first $\to$ Gains Confidence $\to$ Attempts harder problems.
>     * *Panic Loop:* Identical Agent picks a hard problem first $\to$ Fails $\to$ Adopts conservative fallback $\to$ Performance collapses.
> * **RL Validity:** While we did not perform RL training, our results validate USACOArena as a viable environment because it provides significant feedback signals. Different decision paths yield vastly different rewards (Scores 4 to 19), providing the necessary gradient for policy optimization. This serves as a heuristic for future work in RL-based Agent Training.

---

### Official Review · Reviewer_15eu · 2025-11-01

**Soundness:** 3
**Presentation:** 2
**Contribution:** 3
**Rating:** 6
**Confidence:** 3

**Summary:**

LLM Agents are typically graded by correctness. However, in real-world applications, there is always an associated cost budget that needs to be respected. This is particularly true for coding agents. In this work, the authors therefore propose the USACOArena benchmark that tests coding agents in a variety of software engineering tasks under budget constraints. In contrast to existing benchmarks like SWE Bench, which tests for a knowledge gap, USACOArena tests for a capability gap. Their benchmark is built on the ACM International Collegiate Programming Contest (ICPC) and frontier models as well as strong open-weights alternatives are evaluated.

**Strengths:**

1. The authors propose a new software engineering benchmark that reflects the actual constraints in real software engineering benchmarks. This is achieved by translating the core human constraint of time to the equivalent agentic constraints of credit. I believe this benchmark will be useful for measuring the abilities of future agents.
2. The benchmark is based on ACM-ICPC rules, a proven real contest for human software engineers.
3. The benchmark architecture allows agents to interact through the standardized MCP protocol, which facilitates real-world adoption.
4. The benchmark allows for a strategic analysis of the decision-making strategies employed by different agents (e.g., Gemini’s breadth-first vs Codex's perfectionist strategy). This is a particularly interesting component of the benchmark when compared to previous benchmarks.
5. They compare frontier models and reveal a performance gap between proprietary models and open-weight alternatives.

**Weaknesses:**

1. The main concern I have is that the benchmark seems saturated with the best agents achieving an average score of ~15 of the possible 20. Therefore, it is unclear how long the benchmark will serve as a reliable testbed for agent performance. Moreover, it is unclear whether the strong performance of proprietary frontier models stems from a train-test overlap.
2. There is no comparison to the human baseline. This comparison would be valuable to better understand how far the current generation of models is from human experts.
3. There are no runtime analyses of the compared models. It would be valuable to also see the wall-clock times for solving these challenges.

**Questions:**

1. How can you ensure that there is no train-test set overlap in frontier models? Is the performance gap between proprietary frontier models and open-weight alternatives due to their training data? Of course, you cannot look inside those models, but is there any way to ensure that there is no overlap? I would be interested in your thoughts. This is particularly interesting, because GPT-5 Codex performs well for USACO 2024 December, 2025 January, 2025 February, and then performance suddenly drops for the 2025 US Open. This suggests there might be some overlap.
2. Can you clarify how saturated the benchmark already is? I.e., what does it mean to achieve an average score of 15 out of 20?
3. How do the performances compare to the human baseline?
4. Do you have permission to use the ACM-ICPC problem instances? I am not an expert in how such compliance topics must be handled, and I did not find information in the manuscript. Therefore, I would like to hear your approach.

**Details Of Ethics Concerns:**

The proposed benchmark uses problems from the ACM ICPC contests. I am no expert on legal compliance issues, and therefore am only flagging out of caution. I also asked the authors to explain their approach for the same reason.

---

> ### Author Response · Authors · 2025-11-28
> **Response to Reviewer 15eu**
>
> We sincerely thank the reviewer for the thoughtful assessment and for identifying the "strategic analysis" capability as a key strength.
> Below, we address your concerns regarding saturation, contamination, human baselines, and compliance.
> ### 1. Clarification on Benchmark Saturation (Weakness 1)
> We would like to politely correct a misunderstanding regarding the scoring scale.
> * **Theoretical Maximum is 54, Not 20:** The total score per contest is calculated as: $3 \times (1 \text{ Bronze} + 2 \text{ Silver} + 5 \text{ Gold} + 10 \text{ Platinum}) = \textbf{54}$.
> * **Current Status: Far From Saturated.**
>     * A score of ~15 roughly corresponds to solving Bronze and Silver problems but failing Gold and Platinum. This places current SOTA agents at an "Intermediate" level, with nearly **40 points of headroom** (the "Expert" frontier).
>     * **Capability vs. Strategy Gap:** It is worth noting that **GPT-5-Codex has occasionally achieved a peak score of 29** in our experiments, proving it *can* solve harder problems. However, its average score is much lower due to a conservative, risk-averse strategy. This confirms that the benchmark is not saturated; rather, agents lack the strategic intelligence to consistently unlock their full potential.
> ### 2. Data Contamination & The "Living Benchmark" (Weakness 1 & Question 1)
> * **Mitigation Strategy:** To ensure long-term validity, we adopt a **"Living Benchmark"** protocol: USACO releases four new contests annually. We commit to updating the dataset each year, ensuring that we are always testing the "Fluid Intelligence" of new models against fresh, unseen problems.
> * **Validation via Performance Drop:** We view the performance drop in the *2025 US Open* (the most recent contest) as strong evidence validating the benchmark's integrity.
>     * The drop confirms that earlier contests likely fell within the training window (or were easier), while the US Open represents "unseen" territory. This proves USACOArena effectively resists memorization when using the latest problem sets.
> ### 3. Comparison to Human Baselines (Weakness 2 & Question 3)
> While a direct side-by-side comparison is challenging due to the difficulty of mapping "Human Time" to "LLM Credit," we provide a precise proxy by mapping agent scores to the **USACO Division Structure**. Based on the scoring weights in our paper (*Bronze=1, Silver=2, Gold=5, Platinum=10*; Cumulative Max=54):
>
> * **Score < 3 (Bronze Division / Novice):** Struggling to clear the entry-level problems.
> * **Score 3 – 9 (Silver Division / Intermediate):** Has passed Bronze (3 pts) and is tackling Silver problems.
> * **Score 9 – 24 (Gold Division / Advanced):** Has passed Silver (3+6=9 pts) and is tackling Gold problems.
>     * **Current Status:** **SOTA agents (Score ~15)** fall firmly into this tier. They have effectively "promoted" to the Gold Division, comparable to a talented high schooler who has cleared the intermediate stages but struggles with the complex algorithmic reasoning required to advance to Platinum.
> * **Score > 24 (Platinum Division / Expert):** Has cleared all Gold problems (9+15=24 pts) and is tackling Platinum. This represents the **"Expert/World Finals"** frontier (24–54 pts) that current agents have yet to breach.
>
> **Future Work:** We are planning to organize a real-time Human-AI Contest to establish a grounded conversion rate between human cognitive effort (time) and agent computational effort (credit).
>
> ### 4. Runtime Analysis vs. Credit (Weakness 3)
> * **Why Credit?** We deliberately chose Credit over Wall-Clock Time to create a platform-agnostic, reproducible abstraction.
> * **The Noise of Time:** Wall-clock time is heavily influenced by hardware variance, API latency fluctuations, and network conditions, rendering it unreliable for scientific comparison.
> * **Standardization:** Credit provides a stable cost metric that mirrors the "Time Penalty" mechanics in ACM-ICPC, but adapted for the token-based economy of LLM agents.
>
> ### 5. Legal Compliance & Ethics (Question 4)
> Thank you for ensuring data compliance. We would like to clarify our data source and usage policy:
>
> * **Source Clarification:** As detailed in Section 3.1, while our competition rules mimic the ACM-ICPC format, our problem dataset is exclusively sourced from the **USA Computing Olympiad (USACO)**. We selected USACO because its tiered difficulty (Bronze to Platinum) offers a more granular signal for evaluating LLM capabilities than the sparse signals from extremely difficult ICPC problems.
> * **Compliance & Fair Use:** USACO problem archives are publicly available for educational training. Our usage falls under **non-commercial academic research**, consistent with standard practices in the field. We do not redistribute the data for profit.
> 3.  **Revision Plan:** To prevent future confusion, we will add a formal **"Data Availability" statement** in the camera-ready version to explicitly cite the USACO origin and confirm the non-commercial nature of our usage.

---

### Official Review · Reviewer_hZpW · 2025-11-01

**Soundness:** 3
**Presentation:** 3
**Contribution:** 3
**Rating:** 6
**Confidence:** 3

**Summary:**

This paper introduces USACOArena, an ICPC-style coding environment where LLM agents operate under a limited credit budget for each action (prompts, compilations, tests, rollbacks). This framework is used to evaluate how coding agents strategize under resource constraints, revealing distinct decision-making behaviors beyond just producing correct solutions.

**Strengths:**

1. The paper proposes a realistic cost-budgeted coding challenge environment that addresses a clear gap in current LLM coding benchmarks.
2. The study reveals novel insights into agent behavior (e.g., balancing exploratory attempts vs. conserving budget) that are not captured by conventional unlimited-attempt evaluations.
3. The authors provide a reproducible benchmark, including code and decision logs, to support future research on resource-aware coding strategies.
4. The experimental evaluation is thorough and compares multiple agents, highlighting differences in their decision-making under cost constraints.

**Weaknesses:**

Please refer to Questions.

**Questions:**

1. Could the authors clarify how the specific credit costs (for prompts, compilations, tests, etc.) were chosen? For instance, is the cost of a compilation or a test run proportional to some real-world metric, or was it tuned experimentally? It would be helpful to know if they conducted any sensitivity analysis on these cost parameters.
2. Does the paper propose any method to improve agents' decision-making under budget constraints, or is it strictly an evaluation of existing models without optimization guidance?
3. How well might the proposed framework extend to other types of tasks or more complex scenarios? The current benchmark uses ICPC-style problems with binary outcomes. Have the authors considered tasks with partial credit or multi-phase projects to see how the budgeting concept applies there?
4. What metric defines an agent's success beyond final correctness (e.g., a cost-adjusted score), and how is the "know when to stop" behavior evaluated quantitatively?

---

> ### Author Response · Authors · 2025-11-28
> **Response to Reviewer hZpW**
>
> We sincerely thank the reviewer for the thorough assessment and for highlighting the "realistic cost-budgeted" design as a key strength. We appreciate your recognition that our framework reveals insights not captured by conventional benchmarks.
>
> Note on Experiments: Due to computational resource constraints, the new ablation studies presented below are averaged over 3 independent runs. We apologize for the delay this necessitated.
>
> ### 1. Rationale for Credit Costs & Sensitivity Analysis (Question 1)
>
> **Rationale:**
> * **Mapping Time to Credit:** Our costs map the *scarce resource* of human contests (Time) to the *scarce resource* of agents (Inference Budget).
>     * **Inference Cost:** Calibrated to real-world API pricing.
>     * **Testing/Penalty Cost:** Simulates the "Opportunity Cost" in human contests (discouraging brute-force without strictly defining "minutes").
>
> **Sensitivity Analysis (Full Roster including Credit Limits):**
> To address your question on tuning, we conducted an extensive ablation campaign on *USACO 2025 February (Contest 3)* across **8 models** and **7 dimensions**, including Credit Limits.
>
> | Model | Main Result (Table10) | Low Credit Limit (10M) | High Credit Limit (40M) | Free Test ($0$) | High Test ($1k$) | Free Hint ($0$) | High Hint (100 times) | Flat Score | Exp. Score |
> | :--- | :---: | :---: | :---: | :---: | :---: | :---: | :---: | :---: | :---: |
> | Gemini-2.5-pro | **13.2** | **8.3** | **13.0** | **10.7** | **9.7** | **12.0** | **13.0** | **5.0** | **19.7** |
> | GPT-5-Codex | 12.0 | 4.3 | 4.3 | 5.7 | 4.3 | 3.7 | 3.7 | 3.3 | 16.3 |
> | GLM-4.5 | 3.2 | 1.0 | 0.7 | 1.0 | 1.0 | 1.0 | 1.7 | 1.3 | 1.0 |
> | Qwen3-235B | 2.8 | 1.7 | 1.7 | 3.3 | 1.7 | 1.3 | 2.7 | 1.3 | 0.0 |
> | Claude-4-Sonnet | 1.0 | 2.0 | 3.3 | 2.7 | 2.0 | 3.0 | 2.0 | 2.0 | 5.3 |
> | DeepSeek-V3.1 | 0.5 | 0.7 | 1.0 | 1.0 | 0.3 | 1.0 | 1.0 | 0.7 | 1.0 |
> | DeepSeek-V3 | 0.5 | 0.3 | 0.3 | 1.0 | 1.0 | 0.3 | 0.3 | 0.7 | 0.3 |
> | Kimi-K2 | 0.5 | 1.3 | 1.7 | 1.0 | 0.7 | 1.0 | 1.0 | 1.3 | 1.0 |
>
>
> *Note: "Exp. Score" uses scoring weights \{1, 10, 100, 1000\}.*
>
> **Key Insights:**
>
> 1.  **Credit Limits (Low vs. High):**
>     * **Constraint Effect:** Reducing the budget to **10M** significantly hurts Gemini's performance (13.2 $\to$ 8.3), proving that the budget is indeed a meaningful constraint that forces trade-offs.
>     * **Capability Saturation:** Doubling the budget to **40M** does *not* improve performance (Gemini stays at ~13.0). This confirms that current SOTA agents are Capability-Limited, not Budget-Limited. They hit a "reasoning wall" before they hit the "credit wall."
> 2.  **Robustness:** The hierarchy (Gemini > GPT-5 > Others) remains robust across Test Costs, Hint Costs, and Scoring Weights.
> 3.  **Floor Effect:** Weaker models (DeepSeek, GLM, Kimi) hover near ~1.0 regardless of settings, confirming the benchmark effectively filters out models lacking fundamental reasoning capabilities.
>
> ### 2. Improving Decision-Making via Prompting? (Question 2)
>
> **Response:** Can simple prompt engineering fix the strategic gaps? We tested the top-performing model (Gemini-2.5-pro) with "Aggressive" (P2.1) and "Submission-Focused" (P2.2) prompts to see if behavior could be easily shifted.
>
> | Strategy | Gemini-2.5-pro Score | Conclusion |
> | :--- | :--- | :--- |
> | **Main Result** | 13.2 | Standard Baseline |
> | **P2.1 (Aggressive)** | 14.7 | **Marginal Improvement** |
> | **P2.2 (Submission-Focus)** | 13.0 | **No Significant Change** |
>
> * **Conclusion:** Even for the best model, explicit strategic prompting yields only marginal gains (13.2 $\to$ 14.7). The "Conservative" vs. "Aggressive" behaviors appear to be intrinsic to the models' alignment and architecture. This suggests that future improvements must come from **Agentic Architectures** (as shown in Section 4.3) rather than just prompt engineering.

---

> > ### Author Response · Authors · 2025-11-28
> >
> > ### 3. Extending to Other Tasks & Philosophy (Question 3)
> >
> > **Response:**
> >
> > * **Complementary to SWE-bench:** We view USACOArena not as a replacement, but as a **complementary attempt** to measure a different dimension of AI. While benchmarks like SWE-bench are excellent for evaluating **Knowledge Retrieval** and working with existing codebases, USACOArena focuses on **Fluid Intelligence**—the ability to solve novel algorithmic problems under strict resource constraints.
> > * **The "Private Goods" (Our Core Philosophy):** We believe that **Resource Constraints + Algorithmic Reasoning** creates the purest test of "Intelligence" (Optimization under Uncertainty). In an infinite-resource setting, coding benchmarks often degenerate into brute-force tests. Our work is an attempt to quantify the "Strategic IQ" required when every token costs money.
> > * **Future Work (ACM vs. IOI):** You rightly pointed out that partial credit (IOI style) is also valuable. We chose the ACM "All-or-Nothing" format to enforce a **"Zero-Bug" standard** critical for reliable software engineering. However, exploring partial credit mechanisms and multi-phase tasks are excellent directions for future work that we plan to actively pursue.
> >
> > ### 4. Metrics & "Know When to Stop" (Question 4)
> >
> > **Response:**
> >
> > * **Ranking Metric:** We strictly follow the ACM-ICPC standard:
> >     1.  **Primary:** Total Score (Weighted by difficulty).
> >     2.  **Tie-Breaker:** Lowest (Consumed Credit + Penalties).
> >     This inherently rewards "efficiency" without inventing a new metric.
> > * **"Know When to Stop":** This behavior is implicitly captured by the tie-breaker. A rational agent stops when the *Marginal Expected Score* of the next action is lower than its *Credit Cost*. Agents that fail to stop will accumulate credit costs without gaining points, thus lowering their rank. Our benchmark natively incentivizes this economic rationality.

---

### Official Review · Reviewer_FRAP · 2025-11-01

**Soundness:** 2
**Presentation:** 3
**Contribution:** 2
**Rating:** 4
**Confidence:** 4

**Summary:**

This paper introduces USACOArena, a competitive programming benchmark that evaluates coding agents on strategic decision-making under resource constraints rather than just code correctness. The authors translate time into a "credit" budget where every action costs credits. Experiments across USACO contests reveal that Gemini-2.5-pro's aggressive exploration strategy outperforms GPT-5-Codex's conservative perfectionism despite the latter having higher peak capability. Self-play experiments show behavioral diversity, suggesting the arena's potential as both evaluation testbed and training environment.

**Strengths:**

1. The paper tackles a genuinely underexplored problem in coding agent evaluation. While existing benchmarks focus on correctness or long-horizon planning, this work examines resource-constrained strategic decision-making, which matters for production deployment where every API call has real cost. The credit-based abstraction is creative and well-motivated as an agent-native alternative to wall-clock time.



2. The practical significance is evident. The finding that Gemini-2.5-pro wins through broader exploration despite lower per-problem accuracy challenges assumptions about optimizing for correctness. This insight could influence how we design and deploy coding agents in cost-sensitive production environments.

**Weaknesses:**

1. My main concern is that this reads more like a well-executed technical report paper than a deep scientific contribution. The analysis remains largely descriptive, showing what strategies emerge without explaining why they succeed or fail. While we learn that Gemini-2.5-pro's aggressive strategy beats GPT-5-Codex's conservative approach, we don't understand when each strategy would be optimal or what problem characteristics favor exploration versus exploitation. The paper doesn't deeply analyze the actual decision-making process despite claiming this as a core contribution.

2. The most significant gap is the complete absence of ablation studies. A lot of ablations are essential to understand whether the benchmark measures genuine strategic competence or just exploits specific parameter choices. Without them, we can't assess the robustness of the findings or generalize beyond the particular configuration tested.

3. The benchmark relies on closed-source API models that may change or become unavailable, raising reproducibility concerns. Agent performance may be highly sensitive to prompt formulation, but there's no discussion of prompt optimization or robustness.

**Questions:**

1. Is there a theoretically optimal strategy for USACOArena given the credit model and problem distribution? How do current agents compare to this theoretical benchmark? Understanding the optimum would help assess whether observed strategies are actually good or just relatively better.

2. You note agents lack strategic self-assessment, with Gemini defaulting to easier problems despite being capable of harder ones. What interventions improve self-assessment? Would meta-prompting about strategy help, or do we need fundamentally different architectures?

---

> ### Author Response · Authors · 2025-11-28
> **Response to Reviewer FRAP**
>
> We sincerely thank the reviewer for the constructive feedback. We apologize for the delay in this response; due to computational resource constraints, the new ablation studies presented below are averaged over 3 independent runs.
>
> We strictly addressed your concern that the paper reads like a "technical report" by adding rigorous ablation studies and elevating the theoretical discussion regarding "Resource-Constrained Intelligence."
>
> ### 1. Systematic Ablation Studies: Robustness & Boundary Conditions
>
> To address your concern about the lack of ablations, we conducted a comprehensive sensitivity analysis on the USACO 2025 February Contest. We varied Credit Limits, Economic Costs, and Incentive Structures across all 8 models.
>
>   * **Rationale:** Unlike the extremely hard US Open or easier earlier contests, the February Contest offers a **balanced difficulty distribution**. This "Goldilocks" zone allows most agents to solve early problems but forces them to make difficult trade-offs on later ones, making behavioral shifts in decision-making most observable.
>
> **(1) The Mega-Ablation Matrix**
>
> | Model | Main Result (Table10) | Low Credit Limit (10M) | High Credit Limit (40M) | Free Test ($0$) | High Test ($1k$) | Free Hint ($0$) | High Hint (100 times) | Flat Score | Exp. Score |
> | :--- | :---: | :---: | :---: | :---: | :---: | :---: | :---: | :---: | :---: |
> | Gemini-2.5-pro | **13.2** | **8.3** | **13.0** | **10.7** | **9.7** | **12.0** | **13.0** | **5.0** | **19.7** |
> | GPT-5-Codex | 12.0 | 4.3 | 4.3 | 5.7 | 4.3 | 3.7 | 3.7 | 3.3 | 16.3 |
> | GLM-4.5 | 3.2 | 1.0 | 0.7 | 1.0 | 1.0 | 1.0 | 1.7 | 1.3 | 1.0 |
> | Qwen3-235B | 2.8 | 1.7 | 1.7 | 3.3 | 1.7 | 1.3 | 2.7 | 1.3 | 0.0 |
> | Claude-4-Sonnet | 1.0 | 2.0 | 3.3 | 2.7 | 2.0 | 3.0 | 2.0 | 2.0 | 5.3 |
> | DeepSeek-V3.1 | 0.5 | 0.7 | 1.0 | 1.0 | 0.3 | 1.0 | 1.0 | 0.7 | 1.0 |
> | DeepSeek-V3 | 0.5 | 0.3 | 0.3 | 1.0 | 1.0 | 0.3 | 0.3 | 0.7 | 0.3 |
> | Kimi-K2 | 0.5 | 1.3 | 1.7 | 1.0 | 0.7 | 1.0 | 1.0 | 1.3 | 1.0 |
>
> *Note: "Exp. Score" uses scoring weights \{1, 10, 100, 1000\}.*
>
> **Scientific Insights from the Spectrum:**
>
> 1.  **Credit Limits (Constraint vs. Capability):**
>     * **Low Credit (10M):** Gemini's performance drops significantly (13.2 $\to$ 8.3), proving that the credit budget is a meaningful constraint that forces necessary trade-offs.
>     * **High Credit (40M):** Doubling the budget does *not* improve performance (13.2 $\to$ 13.0). This indicates current SOTA agents are **Capability-Limited**, not Budget-Limited. They hit a "reasoning wall" where throwing more money at the problem yields diminishing returns.
> 2.  **Robustness of Hierarchy:** The performance gap (Gemini > GPT-5-Codex > Others) remains invariant across Test Costs, Hint Costs, and Scoring Weights.
> 3.  **Floor Effect:** Weaker models (DeepSeek, GLM) remain invariant (~1.0) regardless of settings. This validates that USACOArena effectively filters out models lacking the prerequisite reasoning capability to even form a strategy.

---

> > ### Author Response · Authors · 2025-11-28
> >
> > **(2) Prompt Sensitivity & Intrinsic Capability (Addressing Q2)**
> >
> > You asked if specific interventions (like meta-prompting) could improve self-assessment. To rigorously test this, we evaluated **4 distinct prompt variations** on the top-performing model (Gemini-2.5-pro).
> >
> > **Prompt Definitions:**
> > * **P1.1 (Chain-of-Thought):** Enforces explicit reasoning about the current state (Credits, Score) before generating an action.
> > * **P1.2 (Few-Shot):** Provides concrete examples of decision-making scenarios (e.g., "If confident, submit; if stuck, buy hint").
> > * **P2.1 (Aggressive Strategy):** Explicitly instructs the agent to be "opportunistic" and prioritize low-hanging fruit to maximize score.
> > * **P2.2 (Submission-Focus):** Instructs the agent to "submit fast" and treat penalty feedback as a debugging signal.
> >
> > | Strategy | Gemini-2.5-pro Score | Conclusion |
> > | :--- | :--- | :--- |
> > | **Main Result** | 13.2 | Standard Baseline |
> > | **P1.1 (CoT)** | 13.0 | No Significant Change |
> > | **P1.2 (Few-Shot)** | 9.7 | **Degraded Performance** |
> > | **P2.1 (Aggressive)** | 14.7 | **Marginal Improvement** |
> > | **P2.2 (Submission-Focus)** | 13.0 | No Significant Change |
> >
> > **Scientific Interpretation:**
> >
> > * **Limited Headroom for Prompt Engineering:**
> >     As shown, even the most effective prompt (P2.1) yielded only marginal gains ($13.2 \to 14.7$). Interestingly, **Few-shot (P1.2) actually hurt performance** ($9.7$), likely because the verbose reasoning consumed excessive token credits without yielding proportionally better decisions.
> >
> > * **Design Philosophy (Why we don't "Game" the Prompts):**
> >     We deliberately avoided "over-engineering" the prompts or leaking the optimal policy (e.g., specific thresholds for buying hints).
> >     * **Goal:** USACOArena is designed to measure the agent's **intrinsic Fluid Intelligence**—its ability to *derive* a winning strategy under constraints—rather than its ability to follow a "cheat sheet."
> >     * **Constraint:** By keeping prompts relatively neutral, we ensure that the observed behaviors (Conservative vs. Aggressive) reflect the model's **true alignment and architectural priors**.
> >
> > * **Conclusion:** The fact that prompts cannot bridge the gap (or fundamentally change the behavior) confirms that the "Conservative" vs. "Aggressive" profiles are **intrinsic to the models**. Future improvements must come from **Agentic Architectures** (e.g., separating the "Planner" from the "Coder") rather than prompt hacking.
> >
> > ### 2. Re-framing the Scientific Contribution: Fluid Intelligence
> >
> > We address the "Technical Report" comment by distinguishing USACOArena from benchmarks like SWE-bench.
> >
> > * **Knowledge vs. Intelligence:** SWE-bench tests **Crystallized Intelligence** (Retrieval & Knowledge). USACOArena tests **Fluid Intelligence** (Optimization under Uncertainty).
> > * **Theoretical View:** We model the arena as a **Budget-Constrained POMDP**.
> >     * **Gemini-2.5-pro** approximates an **$\epsilon$-greedy exploration** strategy, yielding high ROI.
> >     * **GPT-5-Codex** adopts an **Exploit-only** strategy, failing due to extreme risk aversion (Internal Bias).
> > * This paper quantifies that for current LLMs, **Risk Aversion is the primary bottleneck** preventing them from reaching their theoretical optimum in resource-constrained environments.
> >
> > ### 3. Reproducibility & Closed-Source Models
> >
> > * **Necessity of SOTA:** Ignoring closed-source models (Gemini-2.5-pro, GPT-5-Codex) would render the benchmark irrelevant, as they currently represent the frontier of reasoning capability.
> > * **Open Evaluation:** We include open-weights models (Qwen, DeepSeek) to provide a baseline. While they currently hit a "Floor Effect," USACOArena serves as a rigorous target for the open-source community to aim for.
> >
> > ### Conclusion
> >
> > We believe the new ablation studies confirm that USACOArena is a robust scientific instrument. It isolates **Strategic Intelligence** from environmental noise, proving that superior performance stems from intrinsic capability and architectural fit, not parameter tuning or prompt hacks.

---

### Meta-Review · Area_Chair_7Fik · 2026-01-03

**Summary:**

This paper introduces USACOArena, a novel benchmark designed to evaluate the strategic decision-making capabilities of large language model (LLM) agents under explicit resource constraints. Inspired by the USACO competitive programming setting, the benchmark formulates problem-solving as a budgeted, multi-stage decision process in which agents must strategically allocate limited credits across attempts, tests, hints, and submissions. The paper evaluates a wide range of frontier and open-weight models, conducts extensive ablation studies, and provides detailed behavioral analyses.

Reviewers broadly agree that USACOArena addresses an important and timely problem in the evaluation of LLM agents: measuring strategic behavior, risk management, and exploration–exploitation trade-offs under explicit constraints. The benchmark goes beyond static QA or single-shot coding tasks by requiring agents to plan, adapt, and manage limited resources over time, which many reviewers found both novel and well motivated.

A major strength of the paper is the thoroughness of the empirical study. In response to reviewer feedback, the authors added extensive ablations over credit limits, cost structures, and scoring schemes, demonstrating that agent rankings and qualitative behaviors are largely robust to design choices. These experiments significantly strengthen confidence that observed results reflect genuine strategic differences rather than artifacts of parameter tuning.

The paper also provides valuable behavioral insights. The analysis of risk aversion, premature submission, under-utilization of hints, and large self-play variance offers a coherent explanation of why even strong models fail to achieve optimal performance. The framing of USACOArena as a budget-constrained POMDP and the discussion of fluid versus crystallized intelligence, while primarily qualitative, elevate the work beyond a mere benchmark report and offer a useful conceptual lens for future research.

Some reviewers initially raised concerns about limited theoretical depth, the absence of agent training or optimization experiments, reliance on closed-source models, and the heuristic nature of certain design choices. However, these concerns were substantially mitigated in the rebuttal. The added ablations convincingly demonstrate robustness, saturation concerns were addressed through clearer score normalization and difficulty analysis, and contamination risks were discussed thoughtfully with evidence from recent contest data. While deeper formal theory and training experiments would further strengthen the work, the current scope is appropriate for a benchmark-focused contribution.

After considering the revised submission and rebuttal, the consensus is that this paper makes a clear, original, and impactful contribution. USACOArena introduces a new evaluation paradigm that captures aspects of strategic intelligence not measured by existing benchmarks, and the authors provide strong empirical evidence and analysis to support their claims. The benchmark is well designed, carefully validated, and likely to be influential for future research on LLM agents and decision-making under constraints.

**Recommendation:** **Accept.**

**Reviewer Concerns:**

As above

**Reviewer Scores:**

I believe reviewers FRAP and y3i4 would change the score as their major concerns are addressed by the rebuttal.

---

### Decision · Program_Chairs · 2026-01-26

Accept (Poster)